# A deleterious Na$_v$1.1 mutation selectively impairs telencephalic inhibitory neurons derived from Dravet Syndrome patients

Yishan Sun[1,2], Sergiu P Paşca[2,3], Thomas Portmann[2†], Carleton Goold[1,2], Kathleen A Worringer[1], Wendy Guan[1], Karen C Chan[2], Hui Gai[2], Daniel Vogt[4], Ying-Jiun J Chen[4], Rong Mao[2], Karrie Chan[1], John LR Rubenstein[4], Daniel V Madison[5], Joachim Hallmayer[3], Wendy M Froehlich-Santino[3], Jonathan A Bernstein[6], Ricardo E Dolmetsch[1,2*]

[1]Novartis Institutes for BioMedical Research, Cambridge, United States; [2]Department of Neurobiology, Stanford University School of Medicine, Stanford, United States; [3]Department of Psychiatry and Behavioral Sciences, Stanford University School of Medicine, Stanford, United States; [4]Department of Psychiatry, University of California, San Francisco, San Francisco, United States; [5]Department of Molecular and Cellular Physiology, Stanford University School of Medicine, Stanford, United States; [6]Department of Pediatrics, Division of Genetics, Stanford University School of Medicine, Stanford, United States

*For correspondence: ricardo. dolmetsch@novartis.com

Present address: †Circuit Therapeutics, Menlo Park, United States

**Abstract** Dravet Syndrome is an intractable form of childhood epilepsy associated with deleterious mutations in *SCN1A*, the gene encoding neuronal sodium channel Na$_v$1.1. Earlier studies using human induced pluripotent stem cells (iPSCs) have produced mixed results regarding the importance of Na$_v$1.1 in human inhibitory versus excitatory neurons. We studied a Na$_v$1.1 mutation (p.S1328P) identified in a pair of twins with Dravet Syndrome and generated iPSC-derived neurons from these patients. Characterization of the mutant channel revealed a decrease in current amplitude and hypersensitivity to steady-state inactivation. We then differentiated Dravet-Syndrome and control iPSCs into telencephalic excitatory neurons or medial ganglionic eminence (MGE)-like inhibitory neurons. Dravet inhibitory neurons showed deficits in sodium currents and action potential firing, which were rescued by a Na$_v$1.1 transgene, whereas Dravet excitatory neurons were normal. Our study identifies biophysical impairments underlying a deleterious Na$_v$1.1 mutation and supports the hypothesis that Dravet Syndrome arises from defective inhibitory neurons.

## Introduction

Dravet Syndrome, initially described by Dr. Charlotte Dravet in the 1970s, represents one of the most severe forms of intractable childhood epilepsy (*Dravet et al., 2011*). Seizures in Dravet Syndrome appear early in infancy, progress throughout childhood, and are often accompanied by developmental delays in language, motor function, learning and social skills (*Dravet, 2011*). Heterozygous loss-of-function mutations in the *SCN1A* gene are the most common genetic basis for Dravet Syndrome (*Catterall et al., 2010*; *Escayg and Goldin, 2010*). *SCN1A* encodes the voltage-gated sodium channel Na$_v$1.1, which contributes the fast depolarization of neuronal membranes during an action potential. Loss of one copy of Na$_v$1.1 might be expected to reduce neuronal excitability, but paradoxically the genetic perturbation causes epilepsy. Initial insights into the neurobiological

function of Na$_v$1.1 came from studies of a mouse model of Dravet Syndrome, where the loss of one copy of *Scn1a* impaired the excitability of hippocampal inhibitory interneurons but not excitatory neurons (*Yu et al., 2006*). The disproportionate contribution of Na$_v$1.1 to the excitability of inhibitory interneurons in the hippocampus, cortex, cerebellum and thalamus was confirmed by additional studies (*Cheah et al., 2012*; *Dutton et al., 2012*; *Kalume et al., 2007*; *Martin et al., 2010*; *Ogiwara et al., 2007, 2013*; *Tai et al., 2014*; *Tsai et al., 2015*), but the effect of the Na$_v$1.1 mutation in the mouse is dependent on the genetic background with seizures occurring in the C57/B6 background but absent in the 129 background (*Mistry et al., 2014*; *Ogiwara et al., 2007*; *Rubinstein et al., 2015*; *Tsai et al., 2015*; *Yu et al., 2006*).

The effect of the genetic background on the mouse phenotype has highlighted the importance of characterizing the cellular effects of Dravet mutations in a human genetic context. Until recently, experiments on human neurons required the use of tissues obtained following surgeries. However, the development of induced pluripotent stem cell (iPSC) technology (*Takahashi et al., 2007*) has made it possible to generate neurons from patients with disease-causing mutations. Several research groups have used iPSCs to study the pathophysiology of Dravet Syndrome. Two studies demonstrated impairment of action potential firing in patient iPSC-derived inhibitory neurons without examining the excitatory neurons (*Higurashi et al., 2013*; *Liu et al., 2016*). Two other studies reported elevated excitability and increased sodium currents in patient-derived excitatory neurons alone (*Jiao et al., 2013*) or both excitatory neurons and inhibitory neurons (*Liu et al., 2013*). The precise reason for differences between the studies is unclear; however, differences in the generation of specific neuronal cell types likely contribute to the discrepancy. A side-by-side comparison of inhibitory neurons and excitatory neurons derived from Dravet and control iPSCs, including detailed cellular characterization of the neuronal cell types would help to resolve some of the controversy surrounding these studies.

In this study, we investigated the p.S1328P mutation in Na$_v$1.1 identified in a pair of twins with Dravet syndrome. We characterized the effect of this mutation on the biophysical properties of the channel in human neurons and used patient-derived iPSCs to study the consequence of this mutation on the activity of inhibitory and excitatory telencephalic neurons. We found that the disease-associated mutation impairs both functional expression and channel gating of Na$_v$1.1. We also found that inhibitory neurons from the Dravet patients have reduced sodium currents and a pronounced defect in action potential firing particularly when stimulated by large injections of current. In contrast, both the sodium currents and the excitability of excitatory neurons from Dravet patients were indistinguishable from those of control subjects. The reduced excitability of Dravet inhibitory neurons could be rescued by ectopic expression of Na$_v$1.1, while reducing Na$_v$1.1 levels by RNAi in wild-type inhibitory neurons could phenocopy the Dravet cells. Together these findings suggest that the p.S1328P mutation reduces the function of the Na$_v$1.1 channel, which selectively impairs the activity of inhibitory neurons in humans.

## Results

### The Na$_v$1.1-p.S1328P mutation impairs functional expression and alters voltage-dependent gating of the channel

We obtained fibroblasts from two twins affected with Dravet Syndrome. Sequencing of Na$_v$1.1 coding sequence from these patients revealed a serine to proline mutation at position 1328 (*Figure 1A*; Reference mRNA: GenBank AB093548.1; reference protein: GenBank BAC21101.1). The Serine-1328 residue is located in a voltage-sensing transmembrane segment of the protein, and it is conserved in human and mouse Na$_v$1.1 α subunits. This mutation has previously been described in other individuals with Dravet Syndrome (SCN1A Infobase: http://www.scn1a.info/; SCN1A variant database: http://www.molgen.ua.ac.be/scn1amutations/) but the functional effects of the mutation are not known. To determine whether the Na$_v$1.1-p.S1328P mutation alters the function of the channel we expressed wild type and mutant Na$_v$1.1 channels in human neurons. To distinguish the exogenous from the endogenous channels we introduced a second mutation, p.F383S, that makes the channel resistant to tetrodotoxin (TTX). This allowed us to block the activity of the endogenous channels with TTX (*Figure 1—figure supplement 1*) (*Cestèle et al., 2008, 2013*) to measure the currents carried by the expressed channels in isolation. Expressing Na$_v$1.1 constructs in neurons allowed us to

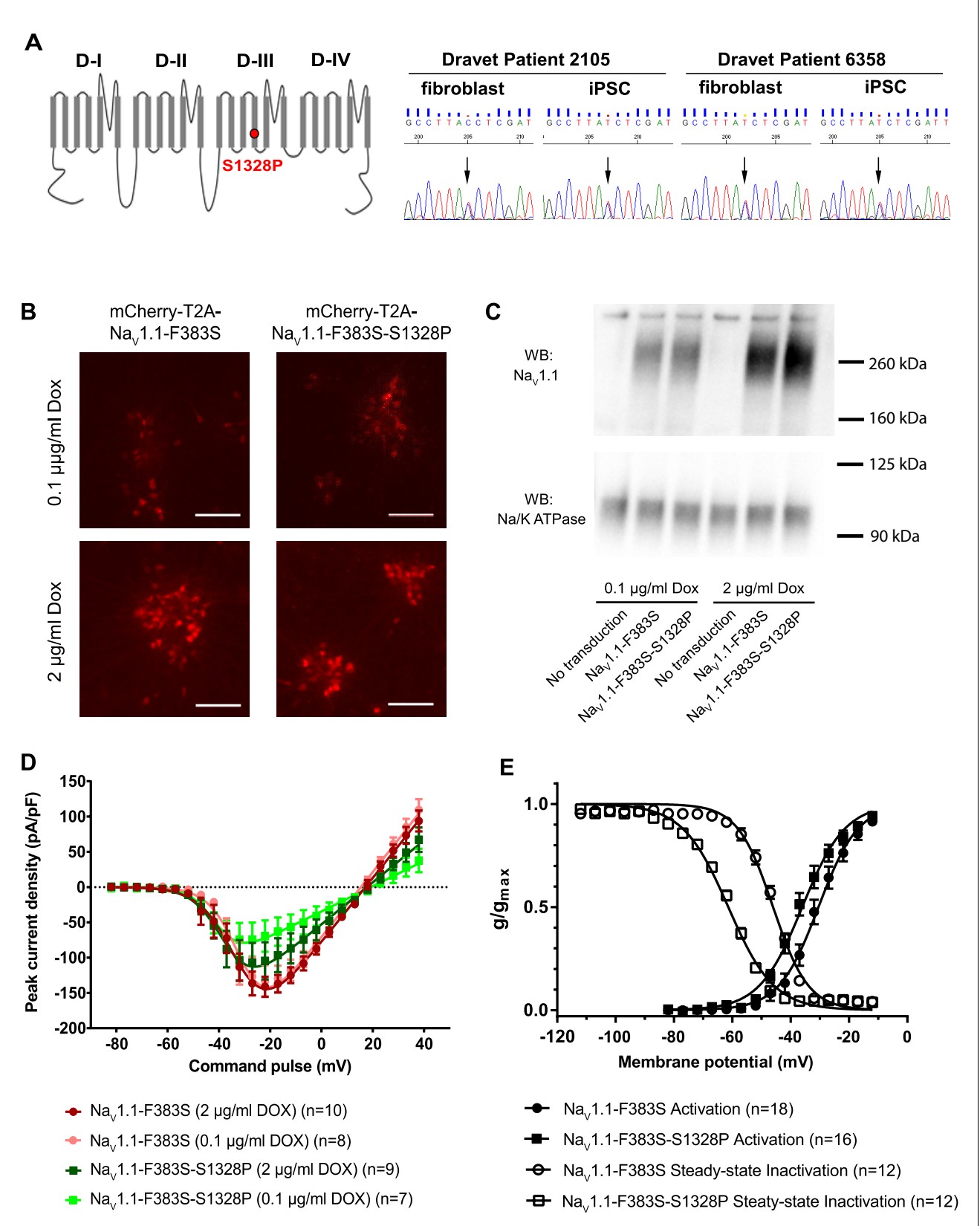

**Figure 1.** Impacts of the Na$_V$1.1-p.S1328P mutation on channel expression and function. (**A**) The p.S1382P mutation is located in Domain III Transmembrane Segment 4 (DIII-S4) of the human Na$_V$1.1 protein. The corresponding DNA variant (c.3982T>C) was confirmed in patient-derived

*Figure 1 continued on next page*

*Figure 1 continued*

fibroblasts and induced pluripotent stem cells (iPSCs) by DNA sequencing. (B) hESCs (H9) were first differentiated into a homogeneous neuronal population via NgN2-induction and then transduced with lentiviral vectors to express mCherry-T2A-Na$_v$1.1-F383S or mCherry-T2A-Na$_v$1.1-F383S-S1328P under the control of a Doxycline (Dox) inducible promoter. The F383S variant makes the channels resistant to TTX. The mCherry reporter labels transduced neurons, and mCherry signal intensity correlated with Doxycline concentrations. (B) Comparing total expression between Na$_v$1.1-F383S and Na$_v$1.1-F383S-S1328P across two Dox concentrations by western blotting (WB). 12.5 µg of total membrane protein was loaded per sample lane, and the ubiquitous membrane protein Na/K ATPase was used as a sample loading control. Na$_v$1.1 was not detected in untransduced NgN2 neurons. There is a non-specific high-molecular-weight band in all lanes in the Na$_v$1.1 blot image. (C) Measurements of TTX-resistant I$_{Na}$ reaveals impaired functional expression of the Na$_v$1.1-F383S-S1328P mutant. By two-way ANOVA using Na$_v$1.1 variants as factor 1 and Dox concentration as factor 2, F $(1, 30)$ = 5.135 and p = 0.0308 between the peak amplitude of Na$_v$1.1-F383S and Na$_v$1.1-F383S-S1328P. See *Table 1* for detailed analyses. (D) Comparing voltage-dependent activation and steady-state inactivation between Na$_v$1.1-F383S and Na$_v$1.1-F383S-S1328P. See *Table 2* for curve fitting and statistical analyses. All error bars are standard errors of the mean.

The following source data and figure supplements are available for figure 1:

**Source data 1.** I$_{Na}$ values quantified in *Figure 1D*, and g/gmax values quantified in *Figure 1E*.

**Figure supplement 1.** Endogenous voltage-dependent sodium currents (I$_{Na}$) of NgN2-induced human H9 neurons were completely blocked by 1-µM tetrodotoxin (TTX).

**Figure supplement 2.** qPCR validation of the expression of endogenous pluripotency genes and silencing of episomal vectors in the Dravet-Syndrome iPSC lines.

**Figure supplement 3.** Further characterization of the Dravet-Syndrome iPSC lines.

**Figure supplement 4.** Characterization of control iPSC lines.

examine the function of the mutant channel in cells that express the correct auxiliary subunits and other molecular partners that regulate channel gating. We made two lentiviruses: mCherry-T2A-Na$_v$1.1-F383S that expresses the engineered control protein and mCherry from a bisystronic message (*Donnelly et al., 2001*; *Szymczak et al., 2004*), and mCherry-T2A-Na$_v$1.1-F383S-S1328P, which expresses the mutant channel and mCherry. Both viruses contained a Doxycline (Dox)-regulated promoter that allowed us to control the expression of the constructs. We transduced the lentiviral vectors into human neurons derived from the human ESC line H9 that expresses the transcription factor from the NgN2 (*Figure 1B*). NgN2-expression generates highly homogeneous and rapidly maturing human excitatory neurons in easily scalable quantity (*Zhang et al., 2013*).

We first used Western blotting to measure the expression levels of control and mutant channels. We found that Na$_v$1.1-F383S and Na$_v$1.1-F383S-S1328P were expressed at similar levels in NgN2 neurons following treatment with either 0.1 µg/ml or 2 µg/ml Dox in the culture media (*Figure 1C*), suggesting that the mutation does not affect the expression of the channel. We then used whole-cell patch clamp to measure the TTX-resistant sodium currents. We found that Na$_v$1.1-F383S-S1328P produces currents that are significantly smaller than those of control channels when the cells are treated with both 0.1 µg/ml and 2 µg/ml Dox (*Figure 1D* and *Table 1*). This suggests that the S1328P either affects the number of channels on the cell surface or the activity of the Na$_v$1.1 channel. The trend that the current observed in the presence of 2 µg/ml Dox was larger than that observed in the presence of 0.1 µg/ml Dox in the cells expressing the mutant channels but not the control channels suggests that there is a defect in trafficking the channel to the cell surfaces (*Figure 1D* and *Table 1*). We also examined the biophysical properties of the Na$_v$1.1-F383S-S1328P channels that do make it to the cell surface. The activation voltage (V$_{1/2}$) was hyperpolarized by around 5 mV, while the steady-state inactivation voltage (V$_{1/2}'$) showed a hyperpolarizing shift of around 15 mV (*Figure 1E* and *Table 2*). This increased sensitivity for activation is likely to enhance the activity of the channel in neurons slightly, however, the much larger increase in steady-state inactivation offsets this effect and probably results in a net loss of function by reducing the fraction of channels that are available to open under prolonged current injections.

**Table 1.** Peak current amplitude of $Na_v1.1$-F383S and $Na_v1.1$-F383S-S1328P expressed in human NgN2 neurons.

| Mean (95% CI) | $Na_v1.1$-F383S | | $Na_v1.1$-F383S-S1328P | |
| --- | --- | --- | --- | --- |
| | 2 µg/ml Dox (10 cells) | 0.1 µg/ml Dox (8 cells) | 2 µg/ml Dox (9 cells) | 0.1 µg/ml Dox (7 cells) |
| $I_{max}$ (pA/pF) | 152.4 (114.4, 190.3) | 141.7 (83.04, 200.4) | 109.6 (44.94, 174.2) | 77.03 (17.77, 136.3) |

$I_{max}$ is the maximal inward current detected in a neuron and normalized to the whole cell capacitance. By two-way ANOVA using $Na_v1.1$ variants as factor 1 and Dox concentration as factor 2, $F_{(1, 30)}$ = 5.135 and p = 0.0308 for factor 1.

Source data 1. The peak $I_{Na}$ amplitude for each neuron described in *Table 1*.

## Generation and characterization of iPSCs from two individuals with Dravet syndrome

We next generated six iPSC lines from the Dravet Syndrome twins carrying the heterozygous p. S1328P mutation in $Na_v1.1$ (*Figure 1A*). We converted skin fibroblasts from the patients into iPSCs using episomal vectors expressing SOX2, OCT3/4, L-MYC, KLF4, LIN28, and p53-shRNA as described previously (*Okita et al., 2011*). We used RT-PCR to verify that the reprogramming vectors were lost in most iPSC cell lines after the 15th passage (*Figure 1—figure supplement 2*). The iPSC lines expressed the pluripotency markers NANOG, TRA-2-49, OCT3/4 SOX2, and LIN28 (*Figure 1—figure supplements 2* and *3*). Four of the iPSC lines responded well to neural differentiation, which was evident by the formation of neural rosettes (*Figure 1—figure supplement 3*). We examined the genomic integrity of the iPSC lines at early passages (passage six to eight) by identifying copy number variants (CNVs) and compared them with CNVs in the source skin fibroblasts. The iPSCs had between three and sixteen novel CNVs, all of which were less than 1.5 Mb in size and none of which occurred in well annotated pathogenic regions (*Supplementary file 1*). The number of novel CNVs in each cell line was below the median number observed in a previous SNP array based characterization of reprogramming associated CNVs (*Hussein et al., 2011*). We also generated six iPSC lines derived from 4 control subjects, 2 males and 2 females, none of whom had a diagnosis of epilepsy. In addition to these iPSCs lines we also included the human embryonic stem cell line H9 (ESC-H9). The use of the control and Dravet cell lines in various experiments is summarized in *Supplementary file 2*.

## Differentiation of human pluripotent stem cells into telencephalic excitatory neurons and inhibitory neurons

We next differentiated iPSCs /ESCs into telencephalic excitatory neurons and inhibitory neurons in parallel. The excitatory neurons were generated using a protocol we have previously described (*Paşca et al., 2011*) (*Figure 2A* top and *Figure 2—figure supplement 1*). Briefly, iPSC/ESC colonies were dislodged from culture vessels by dispase and grown in suspension for five days in the presence of the BMP pathway inhibitor Dorsomorphin (5 µM) and the TGFβ pathway inhibitor SB431542 (10 µM) to form embryoid bodies (EBs). The EBs were plated on poly-L-ornithine and laminin coated dishes and then treated with neuronal induction media containing recombinant human FGF2 (20 ng/ ml) and EGF (20 ng/ml) for ten days to induce neural rosettes. The rosettes were then isolated mechanically and expanded in suspension for seven days before plating onto glass coverslips for post-mitotic neuronal differentiation. Finally the neurons were dissociated and plated onto a monolayer of rat astrocytes for functional maturation over fifty to seventy days. This protocol generates neural stem cells and progenitors that express FOXG1 and PAX6, two markers that are characteristic of telencephalic identity (*Figure 2C*, top) and cortical neurons that express layer-specific cortical markers including FOXP1, SATB2, CTIP2 and ETV1 (*Paşca et al., 2011*).

To generate telencephalic inhibitory neurons we also started with EB formation using the dual SMAD inhibition protocol (*Figure 2—figure supplement 1*). The EBs were exposed to a combination of patterning molecules (*Figure 2A*, bottom) to generate neural stem cells and progenitors with medial ganglionic eminence (MGE) identity. The MGE is a key ventral telencephalic structure that gives rise to inhibitory neurons. We sequentially applied a low-dose (10 nM, between Day 3 and Day

**Table 2.** Voltage sensitivity of $Na_v1.1$-F383S and $Na_v1.1$-F383S-S1328P expressed in human NgN2 neurons.

| Mean (95% CI) | $Na_v1.1$-F383S | $Na_v1.1$-F383S-S1328P |
|---|---|---|
| Activation $V_{1/2}$ (mV) | −30.22 (−33.66, −26.78) | −35.35 [P1] (−38.75, −31.95) |
| Steady-state inactivation $V_{1/2}'$ (mV) | −46.23 (−48.05, −44.42) | −61.43 [P2] (−62.95, −59.91) |

$V_{1/2}$ and $V_{1/2}'$ were determined per neuron by curve fitting in GraphPad Prism using the equation $g/g_{max} = 1/(1+exp((V-V_{1/2})/a))$ and $g/g_{max} = 1/(1+exp((V- V_{1/2}')/a'))$. p1 = 0.0324 by t-test for $V_{1/2}$ values between $Na_v1.1$-F383S and $Na_v1.1$-F383-S1328P. p2 = $1.6 \times 10^{-12}$ by t-test for $V_{1/2}'$ values between $Na_v1.1$-F383S and $Na_v1.1$-F383-S1328P.

Source data 1. $V_{1/2}$ and $V_{1/2}'$ values for each neuron described in *Table 2*.

8) and high-dose (100 nM, between Day 8 and Day 15) of Smoothened agonist SAG (CAS 364590-63-6), to activate Hedgehog signaling for MGE differentiation (*Danjo et al., 2011*). On Day 7, we treated the cell culture for one day with retinoic acid (RA), which promotes the differentiation of MGE cells partly by enhancing Sonic Hedgehog expression (*Goulburn et al., 2012*), and exposed the cells to FGF2 (10 ng/ml) and IGF1 (10 ng/ml) for the next 8 days to promote the proliferation of nascent MGE cells (*Maroof et al., 2010*). Between Day 15 and Day 35, we cultured the cells in neuronal media supplemented with trophic factors including BDNF (20 ng/ml), NT-3 (20 ng/ml), GDNF (10 ng/ml), cyclic AMP (0.5 mM), BMP4 (20 ng/ml, only between Day 15 and Day 25) and IGF1 (10 ng/ml, only between Day 25 and Day 35). On Day 35, the nascent neurons were dissociated into single cells and plated onto a confluent layer of rat cortical astrocytes for fifty to seventy days, as was described for the excitatory neurons.

## Verifying identity of iPSC-derived inhibitory neurons

The neural stem cells and progenitors that were generated using the ventral telencephalon patterning protocol co-expressed transcription factors NKX2.1, FOXG1 and OLIG2 (*Figure 2B and C*, *Figure 2—figure supplement 2A and B*), which together characterize the MGE region in the human embryonic brain (*Maroof et al., 2013*; *Nicholas et al., 2013*). The cells also expressed LHX6 (*Figure 2B*), an MGE-specific transcription factor acting downstream of NKX2.1, as well as transcription factors GSH2 and DLX2 (*Figure 2B*) that cooperate in the development of ventral telencephalic structures including MGE (*Wonders and Anderson, 2006*). In contrast, PAX6, a transcription factor characteristic of dorsal telencephalic progenitors that form excitatory neurons (*Toresson et al., 2000*; *Yun et al., 2001*), was downregulated in the ventral patterning protocol (*Figure 2B and C*). Finally the protocol led to minimal expression of the transcription factors COUP-TFII and SP8 (*Figure 2B* and *Figure 2—figure supplement 2C*). COUP-TFII is known to be expressed in the caudal ganglionic eminence (CGE), the dorsal portion of lateral ganglionic eminence (dLGE), and the dorsal portion of MGE (dMGE). SP8 is largely co-expressed with COUP-TFII in the dorsal CGE (dCGE) and dLGE but absent from MGE (*Cai et al., 2013*; *Ma et al., 2013*). The strong induction of NKX2.1 and mild induction of COUP-TFII and SP8 suggest that a large fraction of the neural stem cells and progenitors made using this protocol have an MGE identity, while a small fraction of the cells have dLGE and/or CGE identity.

The post-mitotic neurons (Day 15 to Day 90) generated from the ventral telencephalic protocol have many features of cortical inhibitory neurons. Among all neurons that were identified by MAP2 staining, about 30% were co-stained by an anti-GABA antibody (*Figure 3A*), suggesting that they produce the inhibitory neurotransmitter GABA. Among the inhibitory (GABA+) neurons, approximately 50% co-expressed Calretinin, and 2%-8% expressed Somatostatin, and the rest expressed neither marker (*Figure 3B and C*). We did not identify any cells that expressed Parvalbumin, possibly because these cells emerge during postnatal development and require both cell-intrinsic and non-intrinsic factors for specification, migration and maturation (*Okaty et al., 2009*; *Sultan et al., 2013*;

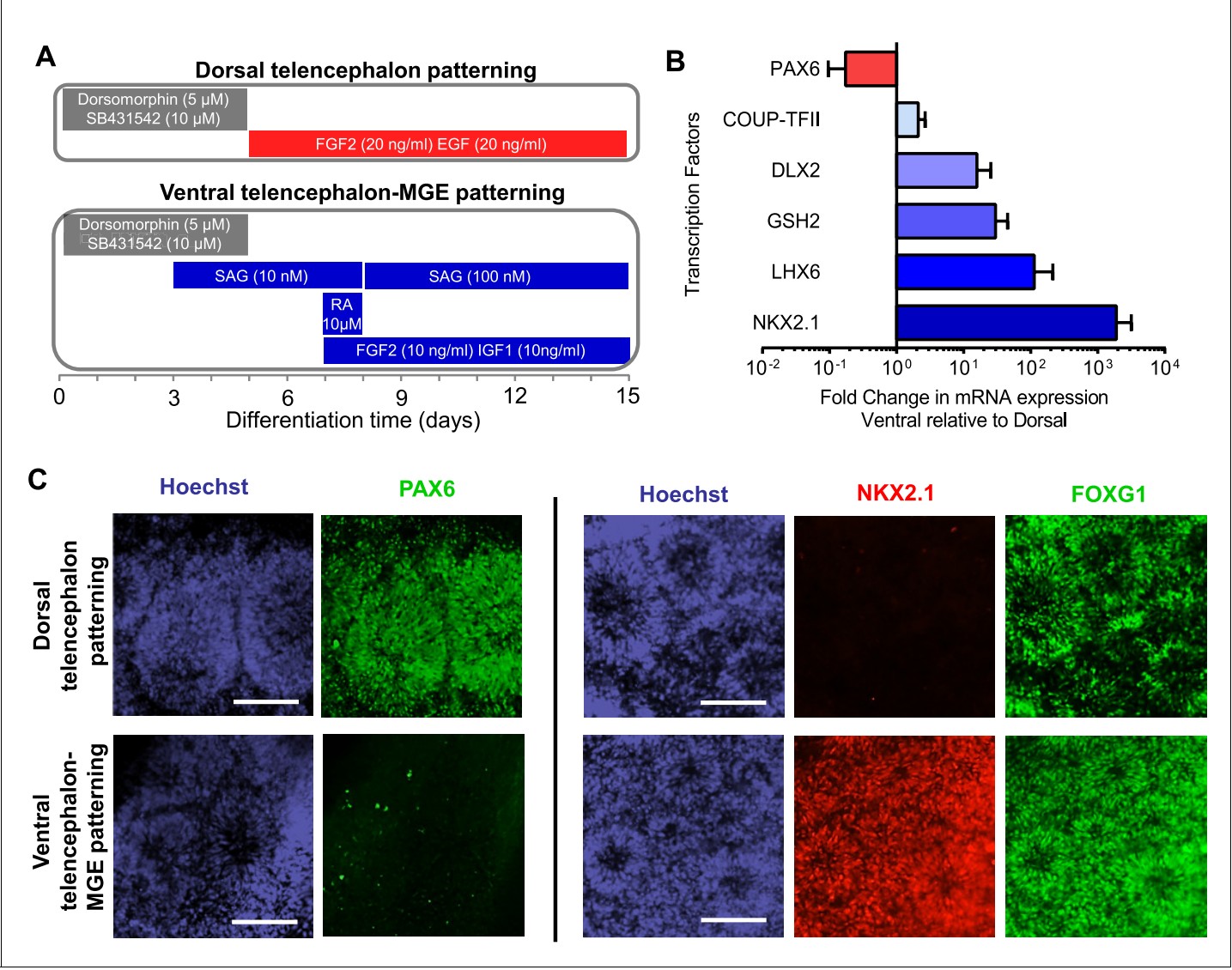

**Figure 2.** Directed differentiation of human ESCs/iPSCs into telencephalic excitatory neurons and inhibitory neurons. (**A**) Extrinsic factors and used to pattern the differentiation of ESCs/iPSCs into neural stem cells and progenitors of either dorsal telencephalic identity (top scheme) or ventral telencephalic and medial ganglionic eminence (MGE) identity (bottom scheme). (**B**) On Day 15 of differentiation, neural stem cells and progenitors were mechanically isolated in the form of neural rosettes. The rosettes were pooled per patterning scheme for RNA isolation and subsequently qRT-PCR analysis of the dorsal telencephalic marker PAX6 and ventral telecephalic markers GSH2, DLX2, LHX6 and NKX2.1. The bar graph represents three experiments in the control hESC line H9, with each experiment pairing a ventral patterning scheme with a dorsal patterning scheme. (**C**) Immunostaining of neural rosettes to examine protein expression of PAX6, NKX2.1, and FOXG1 (a general telencephalic marker) in neural stem cells and progenitors generated from the two patterning schemes, respectively.

The following figure supplements are available for figure 2:

**Figure supplement 1.** A schematic description of the experimental protocol to differentiate human pluripotent stem cells into telencephalic excitatory neurons and inhibitory neurons in parallel.

**Figure supplement 2.** Additional characterization of the ventral telencephalon-MGE induction protocol.

*Wonders and Anderson, 2006*). The expression of Calretinin and low expression of Somatostatin are similar to an earlier report that describes development of telecephalic interneurons using an MGE-biased differentiation protocol (*Nicholas et al., 2013*). However, an alternative scenario could be that the Calretinin-expressing neurons in our current study arise from a population of CGE-like

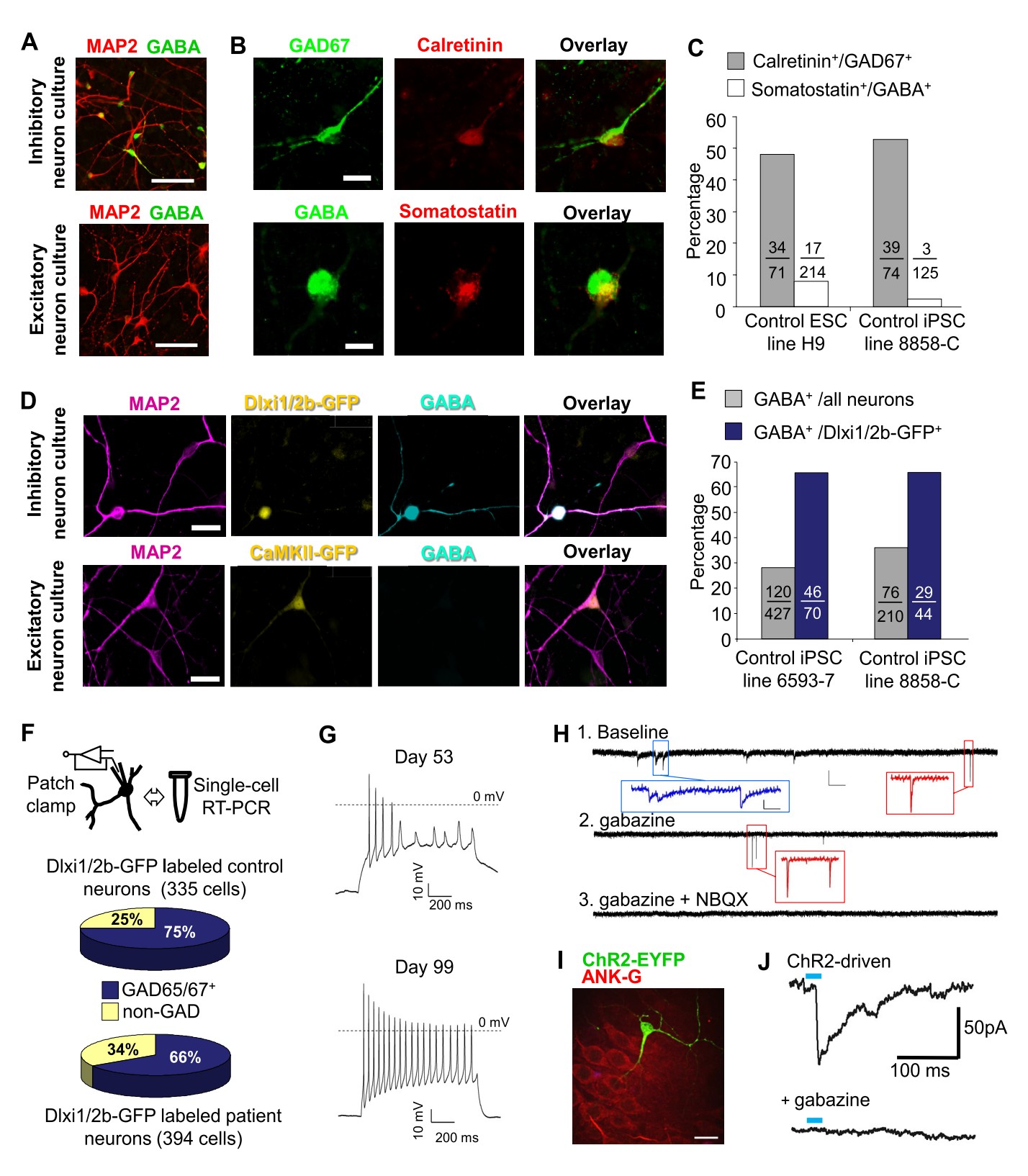

**Figure 3.** Characterization of telencephalic excitatory neurons and inhibitory neurons differentiated from human ESCs/iPSCs. (**A**) An example for the immunostaining of the inhibitory neurotransmitter GABA (in green) and general neuronal marker MAP2 (in red) in differentiated inhibitory neurons (top) and excitatory neurons (bottom). The cell line in the images is hESC-H9. Scale bars, 50 microns. (**B**) Representative images showing a Calretinin[+]

*Figure 3 continued on next page*

*Figure 3 continued*

GAD67$^+$ inhibitory neuron and a Somatostatin$^+$ GABA$^+$ inhibitory neuron. The cell line in the images is hESC-H9. (C) Percentage of Calretinin$^+$ or Somatostatin$^+$ inhibitory neurons from differentiation of hESC-H9 and control iPSC line 8858-C. (D) In inhibitory neuron cultures the expression of lentiviral reporter Dlxi1/2b-GFP overlapped with GABA immunostaining; on the contrary, in excitatory neuron cultures, CaMKII-GFP labeling and GABA immunostaining were mutually exclusive. Scale bars, 20 microns. (E) Around 65% of Dlxi1/2b-GFP$^+$ neurons were stained positive for GABA. Data represent two control iPSC lines 8858-C and 6593–7. (F) By single-cell RT-PCR paired with patch clamp recording, around 70% of Dlxi1/2b-GFP neurons express either GAD65 or GAD67, markers of inhibitory neurons. Data represent all Control (upper pie chart) and Dravet Syndrome (lower pie chart) pluripotent stem cell lines examined in this study. (G) Representative traces showing time-dependent maturation of the action potential firing pattern in inhibitory neurons. The cell line in this example is control iPSC 8858–3. (H) An example for the isolation of Gabazine-sensitive, spontaneous inhibitory synaptic currents (sIPSCs, insets in blue) and NBQX-sensitive, spontaneous excitatory synaptic currents (sEPSCs, insets in red) in cultures of differentiated inhibitory neurons. The detection of sEPSCs reflects the presence of a small fraction of excitatory neurons which likely arise from the low-percentage of PAX6$^+$ progenitors in the ventral telencephalon patterning protocol (*Figure 2C*). The cell line in this example is Dravet iPSC 6358–3. (I) Visualization of a hESC-H9-derived inhibitory neuron engrafted to the CA3 region of a rat hippocampal slice and co-cultured with the slice for 2 months. The human neuron was labeled with the lentiviral vector pLenti-Syn1-ChR2(H134R)-YFP. Rat neurons were visible by Ankyrin-G immunostaining (in red). Scale bar, 25 microns. (J) Under 470-nm light pulses, Gabazine-sensitive inhibitory postsynaptic currents (IPSCs, top trace) were detected in a rat neuron on a hippocampal slice engrafted with ChR2-YFP-expressing, hESC-H9-derived inhibitory neurons. Each trace represents the average of 20 consecutive trials (light pulses).

The following figure supplements are available for figure 3:

**Figure supplement 1.** iPSC-derived MGE-like progenitors and neurons co-express Calretinin during differentiation.

**Figure supplement 2.** Single-cell RT PCR to examine caudal ganglionic eminence-related interneuron markers.

**Figure supplement 3.** Action potential firing patterns of hESC/iPSC-derived inhibitory neurons.

**Figure supplement 4.** Integration of hESC-H9 derived inhibitory neurons into rat hippocampal organotypic slices.

progenitors, as in the adult mammalian cortex where Calretinin expression is mostly restricted to CGE-lineage interneurons (*Wonders and Anderson, 2006*). To distinguish between the two possibilities, we performed double immunostaining for NKX2.1 and Calretinin in differentiating cells before plating them to rat astrocyte feeders. We observed co-expression of NKX2.1 and Calretinin in many cells with progenitor-like morphology and also in a population of cells showing neuronal morphology (*Figure 3—figure supplement 1A*). Furthermore, expression of SP8, an aforementioned transcription factor found in CGE-derived progenitors and neurons, was barely detected in our iPSC-derived inhibitory neurons (*Figure 3—figure supplement 1B and C*). These results are consistent with findings from the differentiating neural rosettes (*Figure 2—figure supplement 2C*) and support the idea that the Calretinin-expressing inhibitory neurons in our study originate from MGE-like progenitors.

Both the excitatory and inhibitory neuronal differentiation protocols generate mixed populations of cells only some of which are mature neurons. We used fluorescent indicators to select the mature neurons for electrophysiological characterization. To identify excitatory neurons we used a lentivirus that expresses EGFP under the control of the CaMKII promoter (*Shcheglovitov et al., 2013*) (*Figure 2—figure supplement 1* and *Figure 3D*). To identify inhibitory neurons we used a lentivirus in which GFP is expressed under the control of an enhancer sequence that is found between the mouse *Dlx1* and *Dlx2* genes (Dlxi1/2b-GFP, *Figure 2—figure supplement 1* and *Figure 3D*). The mouse Dlxi1/2b enhancer is homologous to its counterpart in the human genome and labels MGE-like progenitor cells differentiated from mouse ES cells (*Chen et al., 2013*). Approximately 65% of the Dlxi1/2b-GFP labeled neurons were co-stained by an anti-GABA antibody, indicating they are inhibitory neurons (*Figure 3D and E*). To further identify the cells that were being patch clamped we collected the cell soma after every recording and used single-cell RT-PCR to measure the expression of GAD65 and GAD67. About 70% of the Dlxi1/2b-GFP neurons expressed at least one of the two markers, consistent with the results of the anti-GABA staining and independently of whether the neurons were derived from control or Dravet iPSCs (*Figure 3F*). Furthermore, by single-cell RT-PCR only a small percentage of these GAD67/65-positive inhibitory neurons (0.9% of the control sample group and 5.4% of the Dravet group) expressed 5HT3aR, a serotonin receptor and specific marker

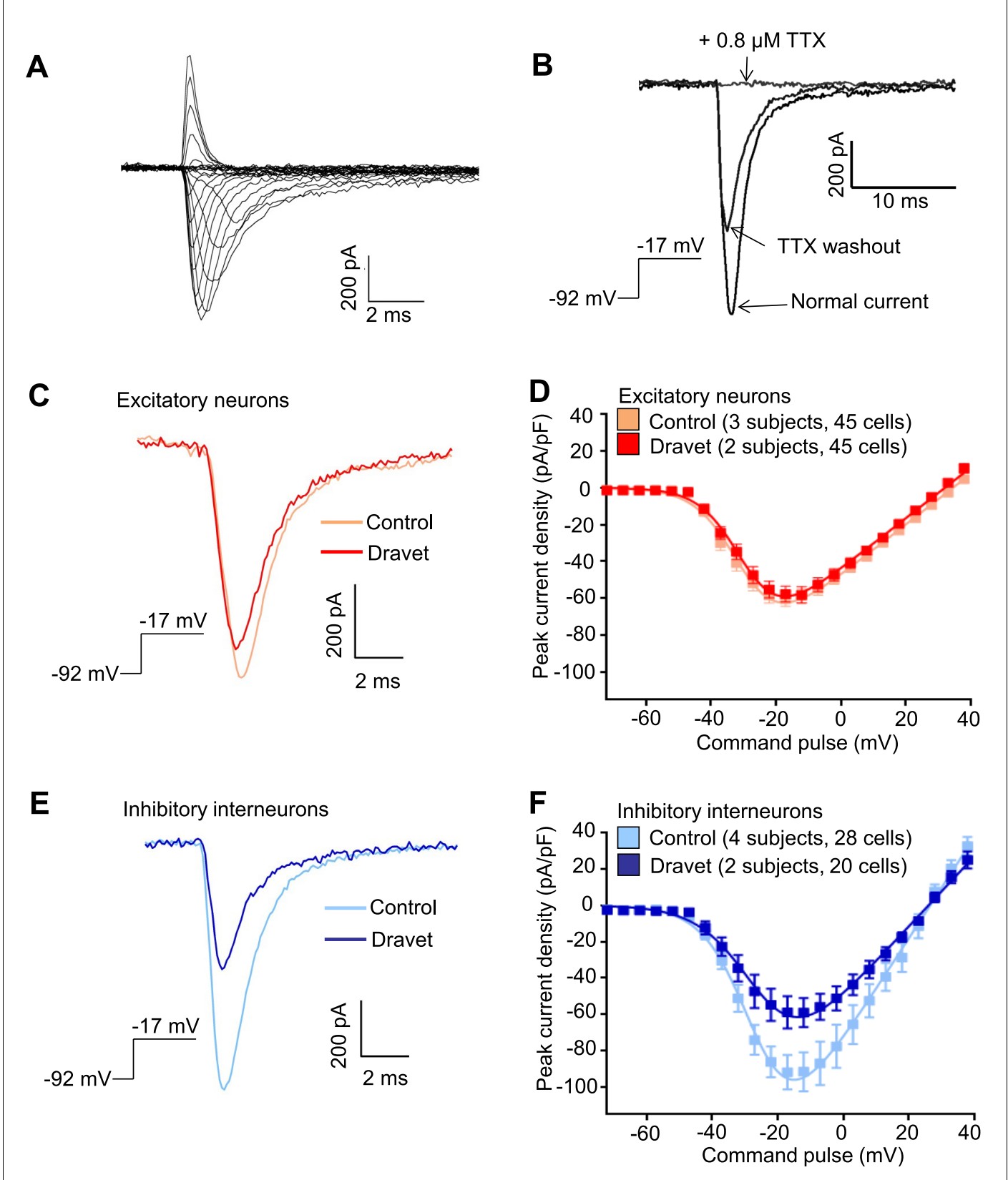

**Figure 4.** Voltage-dependent Na$^+$ currents (I$_{Na}$) were reduced in telencephalic inhibitory neurons but not excitatory neurons derived from Dravet Syndrome iPSCs. (**A**) An example of I$_{Na}$ measured in an inhibitory neuron derived from a control iPSC line. The membrane depolarization steps were set

*Figure 4 continued on next page*

*Figure 4 continued*

between -92 mV and a range of values starting from -72 mV and ending at +38 mV, with 5 mV/step increments. (**B**) Complete blockade of $I_{Na}$ by 0.8 μM TTX. (**C** and **D**) $I_{Na}$ was indistinguishable between control and Dravet excitatory neurons. The example $I_{Na}$ traces in (**C**) were selected to represent the median values of all control excitatory neurons and all Dravet excitatory neurons, respectively. The current trace from the Dravet neuron was scaled relative to the control neuron according to whole-cell capacitance values. Panel (**D**) describes the current-voltage (I-V) relationships in control (n = 45 cells, 3 subjects) and Dravet (n = 45 cells, 2 subjects) excitatory neurons. p = 0.8225 by t-test of the maximal $I_{Na}$ amplitude. (**E** and **F**) $I_{Na}$ amplitude was reduced in Dravet inhibitory neurons relative to control inhibitory neurons. The example $I_{Na}$ traces in (**E**) were selected the same way as in (**C**). Panel (**F**) describes the I-V relationships of control (n = 28 cells, 4 subjects) and Dravet (n = 20 cells, 2 subjects) inhibitory neurons. p = 0.034 by t-test for the maximal $I_{Na}$ amplitude. All error bars are standard errors of the mean. See *Table 3* for detailed statistical analysis.

The following source data and figure supplements are available for figure 4:

**Source data 1.** $I_{Na}$ values quantified in *Figure 4D and F*.

**Figure supplement 1.** Voltage protocols to study sodium currents ($I_{Na}$) in human ESC/iPSC derived neurons.

**Figure supplement 2.** Voltage-dependent $I_{Na}$ steady-state inactivation in iPSC-derived neurons was similar between the control subjects and Dravet Syndrome patients.

**Figure supplement 2—source data 1.** Measurement of $I_{Na}$ steady-state inactivation in each neuron.

for CGE-derived inhibitory interneurons, while more than 60% (for both the control and Dravet groups) expressed Calbindin, which is commonly found in MGE-derived interneurons (*Figure 3—figure supplement 2*). These results are consistent with other characteristics of the neural differentiation protocol and indicate that the cellular composition of the inhibitory neuron pool is dominated by MGE-like rather than CGE-like lineages.

Prior to studying the differences between inhibitory neurons from patients and controls we characterized basic physiological properties of inhibitory neurons from control iPSCs. The Inhibitory neurons fired trains of action potentials upon a 1000-millisecond current injection, and the fraction of cells firing continuous trains increased as the cells matured at least until differentiation day 90 (*Figure 3G*). At this later stage, 73.3% of the inhibitory neurons showed an accommodating pattern of action potential firing; 3.3% had a non-accommodating pattern; a 5% showed a stuttering pattern; and the remaining 18.4% fired at most a few or single action potentials (*Figure 3—figure supplement 3*). We also observed spontaneous inhibitory synaptic currents that were blocked by Gabazine indicating that the cells have functional GABAergic synapses (*Figure 3H*). To determine whether the iPSC-derived inhibitory neurons could integrate into a neural circuit and form functional inhibitory synapses, we infected the cells with a lentivirus expressing Channelrodopsin 2 fused to YFP under the control of the human synapsin-1 promoter (pLenti-Syn1-ChR2(H134R)-YFP) and engrafted them into rat hippocampal organotypic slices. After 8 weeks on the slices, multipolar, ChR2-YFP positive neurons could be observed in the hippocampus (*Figure 3I*; *Figure 3—figure supplement 4A*) containing presynaptic-bouton like structures (*Figure 3—figure supplement 4B*). We used whole-cell patch clamp to measure the currents in the transduced cells and found that illumination with 470 nm light could reliably activate ChR2 and elicit repetitive action potentials (*Figure 3—figure supplement 4C*). Patch clamping of rat neurons surrounding the axons of ChR2-YFP-cells revealed light-driven Gabazine sensitive postsynaptic currents (*Figure 3J*; *Figure 3—figure supplement 4D and E*) indicating that the engrafted human inhibitory neurons formed synapses with the rat neurons. Taken together these experiments suggest that the protocol for specifying inhibitory neurons from iPSCs generated cells that expressed the ventral telecephalic and terminal identity markers, fired repetitive action potentials and formed functional GABAergic synapses.

## Neurophysiological phenotypes of inhibitory and excitatory neurons from patients with Dravet syndrome

We next characterized the physiological properties of inhibitory and excitatory neurons from patients with Dravet syndrome and compared them to cells from individuals without the disease. Because Dravet Syndrome is caused by the loss of one copy of the $Na_v1.1$ channel we first assessed voltage-dependent $Na^+$ currents ($I_{Na}$) in Dravet and control cells. Using whole cell voltage clamp

**Table 3.** Cell-type specific comparisons of $I_{Na}$ activation between control and Dravet iPSC-derived neurons.

| Mean (95% CI) | Excitatory neurons | | Inhibitory neurons | |
| --- | --- | --- | --- | --- |
| | Control (3 subjects, 45 cells) | Dravet (2 subjects, 45 cells) | Control (4 subjects, 28 cells) | Dravet (2 subjects, 20 cells) |
| $I_{max}$ (pA/pF) | 63.62 (55.01, 72.23) | 62.15 [p1] (52.19, 72.11) | 100.5 (74.78, 126.2) | 64.27 [p4] (46.07, 82.47) |
| $V_R$ (mV) | 37.34 (34.51, 40.16) | 34.23 [p2] (31.43, 37.02) | 27.34 (24.32, 30.36) | 28.28 [p5] (23.74, 32.83) |
| $V_{1/2}$ (mV) | −29.89 (−31.58, −28.21) | −28.81 [p3] (−30.31, −27.31) | −27.63 (− 30.06, −25.19) | −25.31 [p6] (−28.98, −21.64) |

$I_{max}$ is the maximal inward current detected in a neuron and normalized to the whole cell capacitance. $V_R$ is the reversal potential for sodium currents. $V_{1/2}$ describes the membrane potential at which 50% of the sodium channels are activated. $V_R$ and $V_{1/2}$ were determined for each neuron by curve fitting in GraphPad Prism using the equation $I = g*(V-V_R)/(1+exp(-0.03937*z*(V-V_{1/2})))$ (see Materials and methods). 95% CI is the 95% confidence interval. p1 = 0.8225, p2 = 0.1291, p3 = 0.3505, in corresponding t-tests between Control and Dravet excitatory neurons. p4 = 0.034, p5 = 0.7259, p6 = 0.2885, in corresponding t-tests between Control and Dravet inhibitory neurons.

Source data 1. Parameters characterizing $I_{Na}$ activation for each neuron described in *Table 3*.

recording (*Figure 4—figure supplement 1A*) under conditions that eliminate contaminating calcium and potassium currents, we observed voltage-dependent and tetrodotoxin (TTX) sensitive sodium currents (*Figure 4A and B*). When we measured $I_{Na}$ in CaMKII-GFP-labeled telencephalic excitatory neurons, there was no significant difference between control and Dravet excitatory neurons in the amplitude (*Figure 4C and D*; *Table 3*), voltage dependence for activation (*Figure 4D* and *Table 3*), or steady-state inactivation profile (*Figure 4—figure supplement 1B*, *Figure 4—figure supplement 2A* and *Table 4*). In contrast, we observed a significant reduction in the $I_{Na}$ amplitude in Dlxi1/2b-GFP-labeled inhibitory neurons from individuals with Dravet Syndrome relative to controls (p=0.034 by t-test; *Figure 4E and F*; *Table 3*). There were no significant differences between control and Dravet inhibitory neurons in the voltage dependence for $I_{Na}$ activation (*Figure 4F*; *Table 3*) or the steady-state inactivation (*Figure 4—figure supplement 2B*; *Table 4*).

Then we assessed the capacity of Dravet and control neurons to fire action potentials in response to injections of a depolarizing current. Both control and Dravet excitatory neurons were able to fire trains of action potentials when injected with up to 60 pA current (*Figure 5A*). There was no statistical difference between control and Dravet excitatory neurons in the maximum frequency of action potential firing ($f_{max}$, *Figure 5B*) or in the quantitative relationship between current injection and the number of action potentials that a cell fired (the dynamic output, *Figure 5C and D*). This is consistent with our previous observations that the $Na_v1.1$ mutation has little or no effect on the sodium current in excitatory neurons.

In contrast when we measured action potential firing in inhibitory neurons we observed a large difference between control and Dravet samples. There was a 40% reduction in the maximal firing frequency of inhibitory Dravet neurons compared to those in controls (*Figure 5E and F*; p = 0.0154 by t-test). At high current injection levels Dravet inhibitory neurons often fired only a few action potentials before entering depolarization block, in contrast to the sustained trains of action potentials observed in the control neurons. Therefore, the input-output relationship of Dravet inhibitory neurons was bell shaped with reduced firing observed at high levels of stimulation (*Figure 5G and H*; between Dravet and control inhibitory neurons p<0.0001 by two way ANOVA, and p<0.05 by post hoc Sidak's multiple comparisons at the two highest stimulation levels). This observation strongly suggests that the loss of one copy of the $Na_v1.1$ channels impairs the ability of inhibitory neurons from Dravet patients to fire at high frequencies.

**Table 4.** Cell-type specific comparisons of $I_{Na}$ steady-state inactivation between control and Dravet iPSC-derived neurons.

| Mean (95% CI) | Excitatory neurons | | Inhibitory neurons | |
| --- | --- | --- | --- | --- |
| | Control (3 subjects, 37 cells) | Dravet (2 subjects, 40 cells) | Control (4 subjects, 25 cells) | Dravet (2 subjects, 19 cells) |
| $V_{1/2}{}'$ (mV) | −44.87 (−46.13, −43.61) | −45.27 [p1] (−46.47, -44.07) | −45.32 (−47.84, −42.79) | −43.31 [p2] (−46.08, −40.55) |

$V_{1/2}{}'$ describes the membrane potential at which 50% of the sodium channels are inactivated. $V_{1/2}{}'$ was determined per neuron by curve fitting in GraphPad Prism using the equation I = 1/(1+exp((V- $V_{1/2}$'/a')) (see Materials and methods). 95% CI is the 95% confidence interval. p1 = 0.6551 by t-test between control and Dravet excitatory neurons. p2 = 0.3035 by t-test between control and Dravet inhibitory neurons.

Source data 1. $V_{1/2}{}'$ values for each neuron described in *Table 4*.

## Enrichment of Na$_v$1.1 in inhibitory neurons correlates with its contribution to neuronal excitability

The simplest explanation for the selective impact of a loss-of-function Na$_v$1.1 mutant on inhibitory neurons in Dravet patients is that the Na$_v$1.1 is preferentially expressed in these cells. We compared the expression of Na$_v$1.1 mRNA between inhibitory neurons and excitatory neurons derived from human iPSCs. When normalized to the expression of NaVβ3 mRNA, which encodes a sodium channel auxiliary subunit uniformly present among subtypes of mouse cortical neurons (*Sugino et al., 2006*), we observed a 1.6–5.6 fold enrichment of Na$_v$1.1 mRNA in cultures of inhibitory neurons relative to excitatory neurons derived from multiple control iPSC lines, Dravet iPSC lines and the hESC line H9 (*Figure 6A*). Using *MAPT* mRNA as an alternative reference for normalization, we obtained a consistent result for the control iPSC line 8858–3 (*Figure 6—figure supplement 1*). The data suggest that Na$_v$1.1 expression is enriched in inhibitory neurons, and the enrichment can be observed within the time frame of human iPSC in vitro differentiation that is much shorter compared to the natural time course of human brain development preceding disease onset. To examine the functional contribution of enriched Na$_v$1.1 expression, we developed lentiviral expression constructs carrying Na$_v$1.1-tRFP-shRNAs (one scramble shRNA sequence and two Na$_v$1.1-targeting shRNA sequences in separate vectors; *Figure 6B*) and a doxycycline-inducible mCherry-T2A-Na$_v$1.1-cDNA (mCherry-T2A tagged wild-type Na$_v$1.1, *Figure 6C*), which served to decrease or increase Na$_v$1.1 expression in neurons, respectively. We first verified that both shRNAs reduced the expression of Na$_v$1.1 mRNA by around 60% (*Figure 6—figure supplement 2*; p<0.005 by Sidak's multiple comparisons following ANOVA). We then transduced the human neurons grown on rat astrocytes and measured $I_{Na}$ currents and membrane excitability in neurons that co-express the lentiviral tracers (tRFP or mCherry) and the cell-type reporters (CaMKII-GFP or Dlxi1/2b-GFP). We found that suppressing Na$_v$1.1 expression using the lenti-shRNAs reduced the $I_{Na}$ amplitude (*Figure 6D* middle) in Dlxi1/2b-GFP$^+$ neurons derived from the control hESC line H9, but not in the CaMKII-GFP$^+$ excitatory neurons (*Figure 6D* left). Conversely, the Na$_v$1.1 lenti-cDNA rescued the $I_{Na}$ amplitude in the Dlxi1/2b-GFP$^+$ neurons derived from Dravet iPSCs (*Figure 6D* right). In parallel to the effects on $I_{Na}$, the Na$_v$1.1-shRNAs impaired action potential firing in control Dlxi1/2b-GFP$^+$ neurons (*Figure 6E*), whereas the cDNA rescued action potential firing in the Dravet Dlxi1/2b-GFP$^+$ neurons (*Figure 6F*). These results indicate that Na$_v$1.1 is a major component of $I_{Na}$ in inhibitory neurons and plays an important role in controlling the excitability of these cells.

## Discussion

Dravet Syndrome is a prototypical form of epileptic encephalopathy caused by loss of function mutations in the Na$_v$1.1 channel. We investigated a Na$_v$1.1 mutation in a pair of twins with Dravet Syndrome and found that it resulted in a decrease in the current carried by the channel but not in a complete loss of function. We studied the effects of this mutation on excitatory and inhibitory

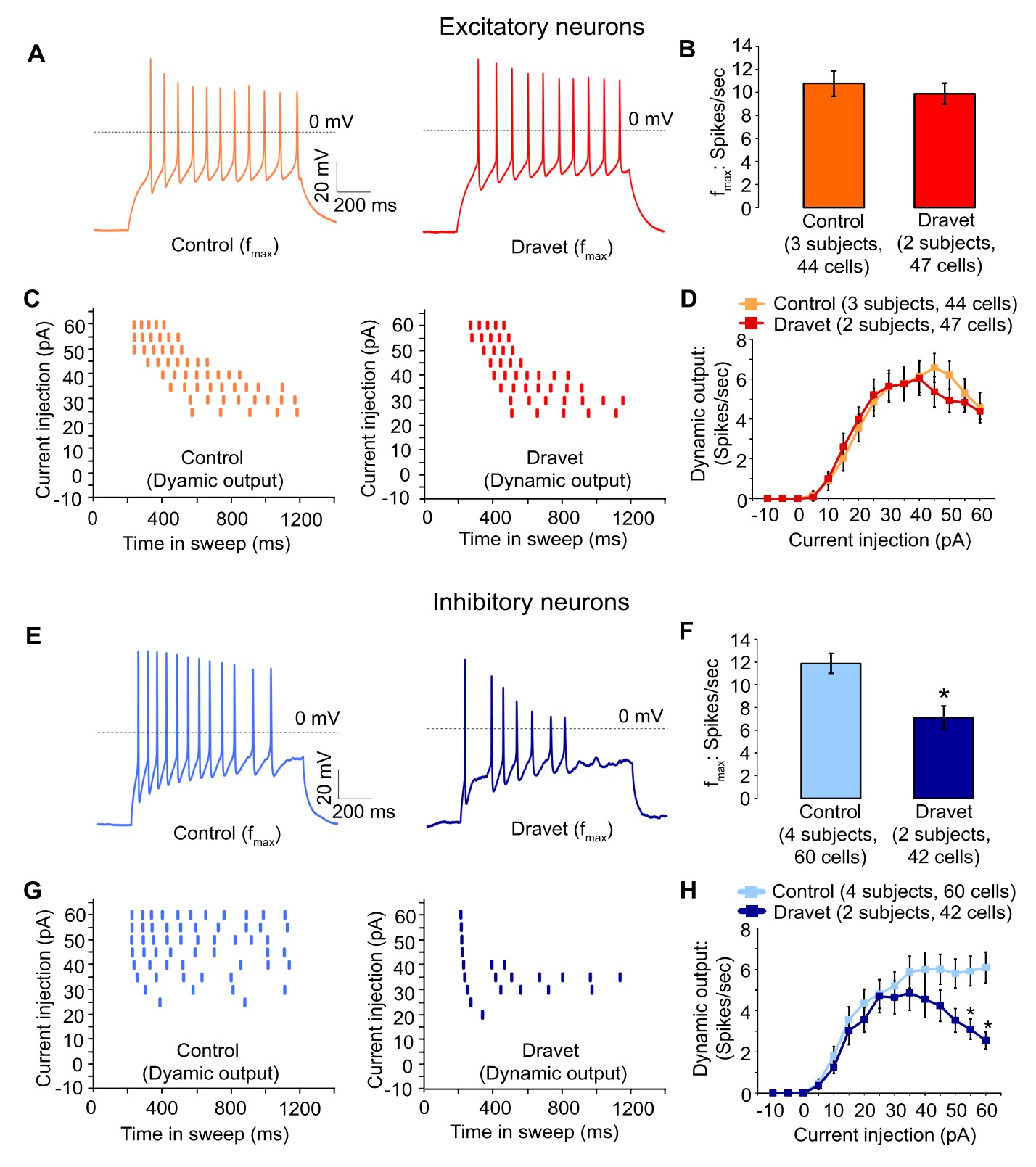

**Figure 5.** Action potential firing were compromised in inhibitory neurons but not excitatory neurons derived from Dravet Syndrome iPSCs. (**A** and **B**) Evaluating the maximal output of action potential firing ($f_{max}$) between Control and Dravet excitatory neurons. The example traces in (**A**) were selected to represent the median $f_{max}$ value of all control excitatory neurons (left panel, n = 44 cells, three subjects) or the median $f_{max}$ of all Dravet excitatory neurons (right panel, n = 47 cells, 2 subjects). In the quantitative comparison (**B**), p = 0.5115 by t-test. (**C** and **D**) Evaluating dynamic output of action

*Figure 5 continued on next page*

*Figure 5 continued*

potential firing between control and Dravet excitatory neurons. Representative raster plots are shown in (**C**). Quantitative input-output relationships are presented in (**D**) as mean ± sem of action potential counts at each current injection level. By two-way ANOVA (factor 1: control versus Dravet neurons; factor 2: current injection levels), $F_{(1, 1335)} = 0.2728$ and $p = 0.6015$ between control and Dravet neurons. (**E** and **F**) Evaluating the maximal frequency of action potential firing between control (n = 60 cells, four subjects) and Dravet (n = 42 cells, two subjects) inhibitory neurons. The example traces in (**E**) were selected the same way as in (**A**). In the quantitative comparison (**F**, marked with asterisk), $p = 0.0006$ by t-test. (**G** and **H**) Evaluating the input-output dynamics of action potential firing between control and Dravet inhibitory neurons. Representative raster plots are shown in (**G**), and quantitative data are summarized in (**H**). By two-way ANOVA, $F_{(1, 1500)} = 21.28$, $p<0.0001$ between Dravet and control neurons. Asterisks at the last two current injection levels indicate $p<0.05$ in *post hoc* Sidak's multiple comparisons between Dravet and control neurons. All error bars are standard errors of the mean.

The following source data is available for figure 5:

**Source data 1.** Action potential numbers quantified in *Figure 5B,D,F and H*.

interneurons by taking great care to generate and characterize human ESC/iPSC-derived telencephalic excitatory neurons and inhibitory neurons in parallel. We found that the p.S1328P mutation affected $I_{Na}$ and action potential firing specifically in patient-derived inhibitory neurons without affecting excitatory neurons. Furthermore, we determined that $Na_v1.1$ is expressed at higher levels in human telencephalic inhibitory neurons than in excitatory neurons, and that it makes a disproportionate contribution to the excitability of these cells. This is consistent with previous findings in mouse models of Dravet syndrome that correlated $Na_v1.1$ expression and function in inhibitory interneurons (*Cheah et al., 2012*; *Dutton et al., 2012*; *Ogiwara et al., 2013*; *Ogiwara et al., 2007*; *Tai et al., 2014*) but differs from some of the previous findings in iPSC models (*Higurashi et al., 2013*; *Jiao et al., 2013*; *Liu et al., 2016*; *Liu et al., 2013*). Specifically, two previous papers have suggested that there is an increase in $I_{Na}$ and neuronal excitability in Dravet neurons (*Jiao et al., 2013*; *Liu et al., 2013*) rather than a selective decrease in $I_{Na}$ and excitability in inhibitory neurons as we have observed here. Two other studies (*Higurashi et al., 2013*; *Liu et al., 2016*) broadly agree with our findings, although the authors only characterized inhibitory neurons, and the developmental origin and trajectory of the inhibitory neurons were not studied.

It is not clear why there is a discrepancy between the various iPSC studies. One possibility is that the different groups studied different sets of mutations which could differentially affect $Na_v1.1$ expression and function. We found that p.S1328P affects the trafficking and gating of $Na_v1.1$ expressed heterologously in neurons. This correlated with the decreased sodium current and excitability in patient-derived inhibitory neurons.

A second possibility underlying the discrepancy between studies is that experimental conditions that were used to measure sodium channel activity and neuronal excitability were different between our work and earlier papers. For example, in two earlier studies the sodium currents were quantified as the inward component of the whole cell current, without pharmacologically eliminating voltage-dependent potassium currents and calcium currents (*Jiao et al., 2013*) or with calcium currents eliminated only (*Liu et al., 2016*). This could confound the results as the measurements of current amplitude, kinetics and voltage dependence could be contaminated by membrane conductance to the non-sodium ions.

A third possibility is that the method we used to generate neurons in this paper produced physiological properties different than those studied in earlier papers. Although in all four prior studies neural differentiation began with the formation of embryoid bodies (EBs), the earlier papers do not provide sufficient information on specific steps of neural differentiation or cell lineage markers for us to determine if the cells are similar to ours. Our neurons resemble inhibitory and excitatory neurons found in the developing human telencephalon, based on the expression of cell-type specific markers in both neural progenitors and neurons. The critical question is whether iPSC-derived neurons in a culture dish fully mimic the neurons in patient brains. While we can't answer this question definitively, it is encouraging that the neurons in our experiments have similar phenotypes compared to those observed in vivo in multiple mouse model of Dravet Syndrome.

Why does loss of $Na_v1.1$ have a greater effect on the excitability of inhibitory neurons relative to excitatory neurons? One possibility might be that excitatory neurons are more effective than inhibitory neurons at upregulating other sodium channels in response to loss of one copy of $Na_v1.1$. Our

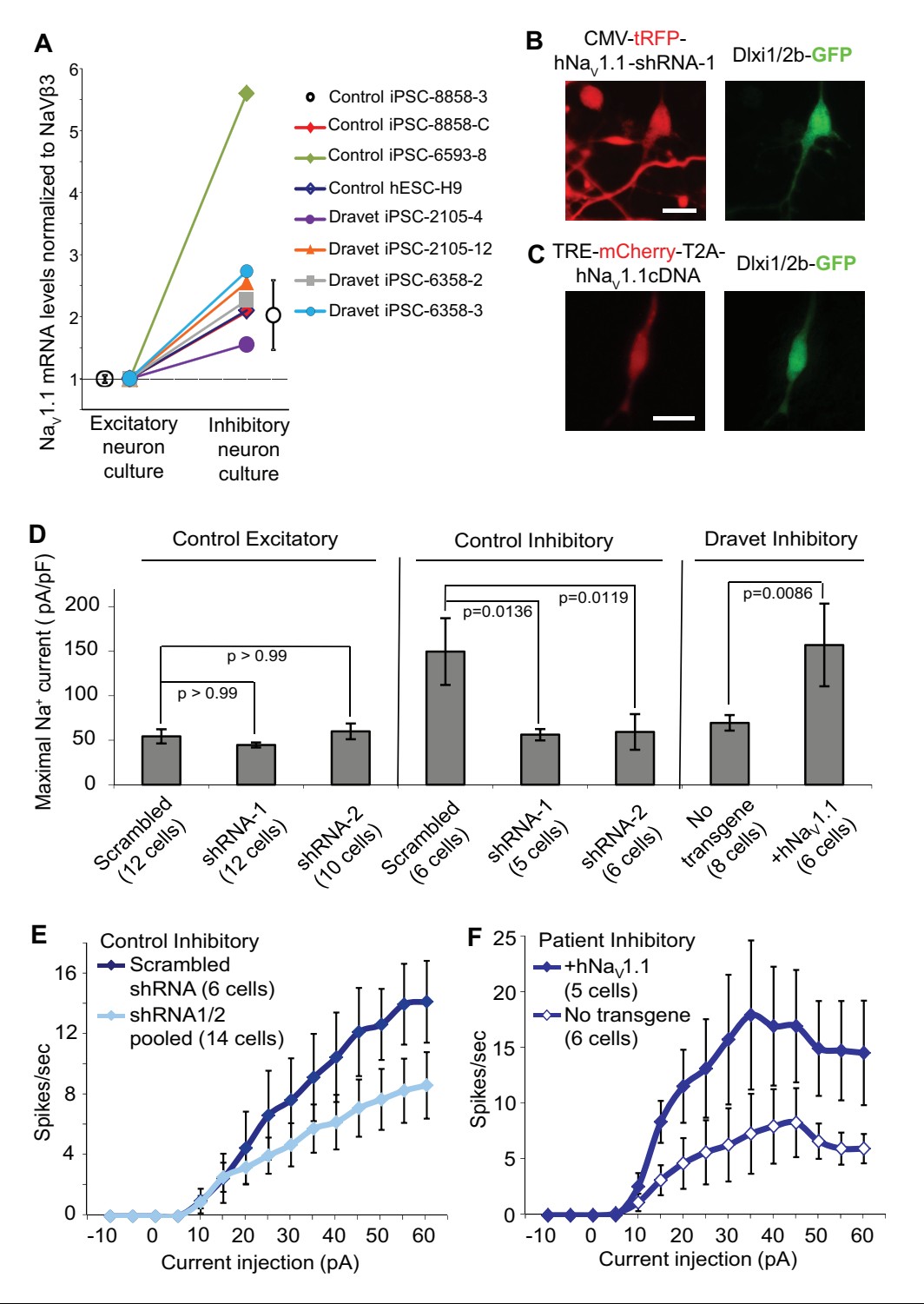

**Figure 6.** Na$_v$1.1 is required for high Na$^+$ current density in inhibitory neurons derived from human pluripotent stem cells. (**A**) Differential expression of Na$_v$1.1 mRNA between cultures of telencephalic excitatory neurons and inhibitory neurons. Quantification of Na$_v$1.1 mRNA was based on real-time PCR and normalization to the auxiliary Na channel subunit NaVβ3. For control iPSC line 8858–3, n = 6 culture wells of excitatory neurons, n = 5 culture wells of inhibitory neurons. For the other cell lines, one culture well of excitatory neurons and one culture well of inhibitory neurons were assessed pairwise. (**B**) Visually identifying Dlxi1/2b-GFP neurons that co-express Na$_v$1.1-shRNAs with a tRFP tracer. Scale bar, 10 microns. Similar tracing was performed for CaMKII-GFP-labeled excitatory neurons co-expressing hNa$_v$1.1-shRNAs. (**C**) Visually identifying Dlxi1/2b-GFP neurons that co-express a doxycycline-inducible Na$_v$1.1 cDNA with a mCherry tracer. Scale bar, 25 microns. (**D**) Na$_v$1.1- shRNA1 and Na$_v$1.1-shRNA2 reduced Na$^+$ currents ($I_{Na}$) in hESC-H9 derived Dlxi1/2b-GFP$^+$ neurons, but not in CaMKII-GFP$^+$ neurons. Conversely, the Na$_v$1.1 cDNA restored the $I_{Na}$ in Dravet Dlxi1/2b-GFP$^+$ neurons (Cell

*Figure 6 continued on next page*

*Figure 6 continued*

line: 6358–2). $F_{(7, 57)}$ = 5.636, p<0.0001 by ANOVA for the entire dataset. Individual p values from *post hoc* Sidak's multiple comparisons are indicated on the graph. (**E**) Na$_v$1.1-shRNAs suppressed action potential firing capacity in hESC-H9 derived Dlxi1/2b-GFP$^+$ neurons. $F_{(1, 270)}$ = 13.24, p = 0.0003 by two-way ANOVA for the effect of Na$_v$1.1-shRNAs on spike number over the range of current injections. (**F**) The Na$_v$1.1 cDNA restored action potential firing capacity in Dlxi1/2b-GFP$^+$ neurons derived from Dravet iPSC line 6358–2. $F_{(1, 135)}$ = 26.69, p<0.0001 by two-way ANOVA for the contribution of Na$_v$1.1 cDNA to spike number over the range of current injections. All error bars are standard errors of the mean.

The following source data and figure supplements are available for figure 6:

**Source data 1.** I$_{Na}$ values quantified in *Figure 6D*, and action potential numbers quantified in *Figure 6E and F*.

**Figure supplement 1.** Differential expression of Nav1.1 mRNA between cultures of telencephalic excitatory neurons and inhibitory neurons derived from the control iPSC line 8858–3.

**Figure supplement 2.** Transduction and RNAi efficiency of two human Nav1.1 lenti-shRNA clones.

data suggests that this is not the case. We observed that human Na$_v$1.1 mRNA was present at higher levels in ESC/iPSC-derived inhibitory neurons than excitatory neurons, and that bi-directional manipulation of Na$_v$1.1 expression level using lentiviral constructs correlated with I$_{Na}$ currents and membrane excitability in the inhibitory neurons. In addition, studies in rodents suggest that cortical inhibitory interneurons express both Na$_v$1.1 and Na$_v$1.6 in their axon initial segments while cortical excitatory neurons express only Na$_v$1.6 channels in this cellular compartment (*Lorincz and Nusser, 2008*; *Ogiwara et al., 2007*). As the axon initial segment plays a key role in the initiation of action potentials, this difference could account for the disproportionate effect of loss of Na$_v$1.1 on inhibitory neuron firing in Dravet Syndrome.

## Materials and methods

### Dravet syndrome iPSC generation and characterization

All procedures and experiments involving primary fibroblasts and iPSCs from Dravet Syndrome patients were carried out at Stanford University after informed consent and under protocols approved by the IRB and SCRO of Stanford University. A clinical skin biopsy procedure was carried out to obtain primary fibroblasts from a pair of twins diagnosed with Dravet Syndrome. The primary fibroblasts were expanded and cryopreserved at the Stanford cytogenetics lab. The primary fibroblast culture medium contains DMEM (base medium, Gibco), FBS (10%, Gibco), NEAA (1%, Gibco), Sodium pyruvate (1%, Gibco), GlutaMax (1%, Gibco), PenStrep (100 units/mL working concentration, Gibco), and β-mercaptoethanol (0.1 mM, Sigma Aldrich). A set of episomal expression vectors encoding SOX2, OCT3/4, KLF4, LIN28, L-MYC and P53-shRNA (*Okita et al., 2011*) were delivered to the fibroblasts using a Basic Primary Fibroblasts Nucleofector Kit (Lonza). The fibroblasts were allowed to recover and then seeded onto mouse DR4 MEF feeders (prepared by Stanford University Transgenic Mouse Facility). The co-culture was maintained for 24 hr in the primary fibroblast medium and later on in standard human ES medium containing DMEM/F12 (base medium, Gibco), KnockOut Serum Replacement (20%, Gibco), GlutaMax (0.5%, Gibco), NEAA (1%, Gibco), Penstrep (100 units/ mL working concentration, Gibco), β-mercaptoethanol (0.1 mM, Sigma Aldrich), and human recombinant FGF2 (10 ng/ml, R&D systems). After 3 weeks, emerging iPS cell colonies were isolated by manual picking and expanded separately as cell lines, which are named in de-identified codes. The established iPSC lines were mycoplasma-free based on tests using the Mycoalert Mycoplasma Detection Kit (Lonza).

Immunocytochemistry using antibodies against the pluripotency markers NANOG (nuclear localization, antibody: R&D Systems AF1997) and TRA-2-49 (cytoplasmic localization, antibody: DSHB Hybridoma Product TRA-2-49/6E, originally developed by P. W. Andrews) were conducted as described before (*Yazawa et al., 2011*). Quantitative RT-PCR was performed to examine the silencing of episomal reprogramming factors and the establishment of corresponding endogenous factors, using PCR primers described in *Supplementary file 3*.

For SNP array assessment of copy number variants (CNVs), cultures of skin fibroblasts and iPSCs were dissociated with Accutase (Gibco), washed three times in PBS, pelleted, and preserved at -80°C. The cell samples were shipped to a commercial service (CapitalBio Corp., Beijing, China) for CNV analysis using the Affymetrix Genome-wide Human SNP Array 6.0. The SNP array contains more than 946,000 probes to detect both known and novel CNVs. Data analysis was based on a copy number polymorphism (CNP) calling algorithm developed by the Broad Institute.

## Control iPSCs

Control iPSC lines NH2-6, 6593–7 and 6593–8 were described in previous studies (*Pasca et al., 2011*; *Yazawa et al., 2011*). Control iPSC lines 8858-C and 8858–3 were generated under informed consent and protocols approved by IRB and SCRO at Stanford University, using skin fibroblasts from healthy donor 8858 and identical reprogramming methods as described for the Dravet Syndrome iPSCs. Basic characterization of these two iPSC lines is described in *Figure 1—figure supplement 4*. Control iPSC line 8402–2 WT9 is a subclone of iPSC line WT-GM08402A-SeV-hiPSC 1, which was originally generated at Novartis Institutes for BioMedical Research, using GW08402 fibroblasts from the Coriell Institute for Medical Research and Sendai-virus-based reprogramming (*Fusaki, et al., 2009*). WT-GM08402A-SeV-hiPSC 1 passed tests for pluripotency gene expression and viral vector removal. In our current study, we verified pluripotency marker expression and confirmed a normal karyotype in the subclone 8402–2 WT9 (*Figure 1—figure supplement 4*). All control iPSC lines were mycoplasma-free based on tests by the Mycoalert Mycoplasma Detection Kit (Lonza).

## Directed differentiation of human pluripotent stem cells (hPSCs) into telencephalic excitatory neurons and inhibitory neurons

An overview of the procedure is presented in *Figure 2A*. First, human ES/iPS cell colonies were detached from DR4 MEF feeders with Dispase and grown as suspension culture for five days to form embryoid bodies (EBs). The EB medium was the standard human ES medium supplemented with SMAD inhibitors Dorsomorphin (5 µM, Sigma Aldrich) and SB431542 (10 µM, Tocris) but without FGF2. Next, EBs were plated for the induction of neural rosettes over 10 days (*Figure 2B*). The neural induction media contained Neurobasal-A (Base medium, Gibco), B-27 without Vitamin-A (2%, Gibco), GlutaMax (1%, Gibco), and Penstrep (100 units/mL working concentration, Gibco). Human recombinant FGF2 (20 ng/ml for dorsal telencephalon induction, 10 ng/ml for ventral telencephalon-MGE induction; R&D systems), EGF (20 ng/ml, EMD Millipore), IGF1 (10 ng/ml, EMD Millipore) and small molecules SAG (10 nM and 100 nM, Enzo Life Sciences, Farmingdale, New York) and retinoic acid (RA) (10 µM, Sigma Aldrich) were added to the neural induction media in specific combinations (*Figure 2A*). Neural rosettes were isolated by mechanical picking and re-plated for neuronal differentiation. For rosettes harvested from the dorsal telencephalon patterning scheme, an extra expansion step was included, where the neural progenitors were allowed to proliferate in suspended neurospheres over seven days (*Figure 2—figure supplement 1*). The neuronal differentiation was divided into an initial stage (20 days) and a secondary, astrocyte- supported stage (~2 months), where the immature human neurons were co-cultured with a rat cortical astrocyte monolayer. The neuronal differentiation media contained Neurobasal-A (Base medium, Gibco), B-27 without Vitamin-A (2%, Gibco), GlutaMax (1%, Gibco), and Penstrep (100 units/mL working concentration, Gibco), recombinant human BDNF (10 ng/ml, PeproTech, Rocky Hill, New Jersey) and recombinant human NT-3 (10 ng/ml, PeproTech). Additional neurotrophic or signaling molecules were used for the initial stage of inhibitory neuron differentiation. Between Differentiation Day 15 and 25 (*Figure 2—figure supplement 1*), we used BDNF (10 ng/ml, extra, PeproTech) and NT-3 (10 ng/ml, extra, PeproTech), GDNF (10 ng/ml, R&D systems), cyclic AMP (0.5 mM, Sigma Aldrich) and BMP4 (20 ng/ml, R&D systems); between Differentiation Day 25 and 35, we used BDNF (10 ng/ml, extra, PeproTech) and NT-3 (10 ng/ml, extra, PeproTech), GDNF (10 ng/ml, R&D systems), cyclic AMP (0.5 mM, Sigma Aldrich) and IGF1 (10 ng/ml, EMD Millipore). The rat cortical astrocytes were prepared from newborn to P2 rat pups, using a procedure modified from a previous publication (*McCarthy and de Vellis, 1980*), according to a lab protocol approved by the administrative panel on laboratory animal care (APLAC) at Stanford University. At the transition to astrocyte-supported neuronal differentiation, human neurons were dissociated with Accutase and plated at the density of 10,000 cells/cm$^2$.

## Rat hippocampal slice culture, engraftment of hPSC-derived inhibitory neurons, and optogenetics in engrafted slices

This procedure was carried out according to a lab protocol approved by the administrative panel on laboratory animal care (APLAC) at Stanford University. The culture of rat hippocampal slices was prepared and maintained according to a standard method (*Hanson, et al., 2010*). In brief, bilateral hippocampi from P6 Sprague Dawley rat pups were isolated and sliced to the thickness of 400 microns. The slices were laid on top of the PTFE membrane of a Millipore millicell culture insert, which was placed in a media filled dish to form an air-liquid interface. CaMKII-ChR2-YFP labeled human inhibitory neurons were concentrated to 35,000 cells/µl, and 1.5 µl (~50,000 cells) were added to the top surface of each slice. The slices were maintained in 5% $CO_2$, at 37°C for 3 days and then at 34°C for two months. The culture media comprises minimum essential medium (MEM, Gibco)/HBSS (Gibco) (2/1 mixed as the base medium), Penstrep (100 units/mL working concentration, Gibco), HEPES (12.5 mM diluted from 1M stock solution, Gibco), heat-inactivated horse serum (25%, Gibco). To drive activity of ChR2, 470-nm light from an LED was pulsed through the 40x immersion lens, under the control of programmed digitizer output. Patch clamp recordings of the grafted human neurons or the host rat neurons were carried out at room temperature and in continuous perfusion of artificial cerebrospinal fluid (ACSF) bubbled with 95% $O_2$, 5% $CO_2$. The $GABA_A$ receptor competitive antagonist Gabazine (SR 95,531 hydrobromide, Tocris) was dissolved in DMSO and used at a final concentration of 10 µM.

## Construction of lentiviral expression vectors

The CaMKII-GFP vector contains a 1.3 kb promoter of the mouse α-calcium/calmodulin-dependent protein kinase II (*Dittgen et al., 2004*). The vector was generated during a previous study in the lab (*Shcheglovitov et al., 2013*). The Dlxi1/2b-GFP vector contains the mouse Dlxi1/2b enhancer, a β-globin minimal promoter, and the EGFP coding sequence in tandem. The vector is a variant of the Dlxi1/2b-β-globin-mCherry vector described previously (*Chen et al., 2013*). The CMV-tRFP-$Na_v$1.1-shRNA vectors were made by replacing the SnaBI-tGFP-XhoI fragment of the pGIPZ-$Na_v$1.1-shRNA vectors (Open Biosystems) by the SnaBI-tRFP-XhoI fragment from pLemir-NS (Open Biosystems). The original pGIPZ-$Na_v$1.1-shRNA vectors were: vector # RHS4346, which carries the scrambled control shRNA; vectorV3LHS_360126, which carries $Na_v$1.1-shRNA-1; vector V3LHS_360129, which carries $Na_v$1.1-shRNA-2. The TRE-mCherry-T2A-h$Na_v$1.1-cDNA vector was made by a two-step cloning procedure. First, a mCherry-Gly-Ser-Gly-T2A-NotI fragment was recombined with the BamHI-SmaI-linearized pTight-ASCL1-N174 plasmid (a kind gift from the Crabtree lab at Stanford University) using the CloneEZ PCR cloning kit (GenSript, Piscataway, New Jersey). The resulting plasmid pTight-mCherry-T2A was linearized by NotI and recombined with the human $Na_v$1.1 wild-type cDNA (encoding the 1997-aa isoform of $Na_v$1.1, under an MTA from GlaxoSmithKline to the Dolmetsch lab at Stanford University), or with $Na_v$1.1-F383S and $Na_v$1.1-F383S-S1328P cDNAs (generated at Novartis) also using the CloneEZ PCR cloning kit. The final construct has these functional elements in tandem: Tetracycline Response Element (TRE), mCherry, T2A (mediating 'self-cleavage' during translation), and human $Na_v$1.1 cDNA.

## Lentivirus preparation and neuronal transduction

Lentiviruses encoding CaMKII-GFP, TRE-mCherry-T2A-$Na_v$1.1-cDNA (wild type), or EF1α-rtTA were prepared at the Stanford University Neuroscience Gene Vector Core, using helper plasmids psPAX2 and pMD2.G. Viral titers (infectious units/ml) were determined by infecting HEK-293 cells and quantitating the amount of integrated provirus by Q-PCR using WPRE-specific probes and primers. Transduction of differentiating human neurons was carried out 4 days before re-plating to rat cortical astrocytes. The CaMKII-GFP virus was used at MOI = 1 (1 infectious unit/cell). The TRE-mCherry-T2A-$Na_v$1.1-cDNA virus and EF1α-rtTA virus were used together at MOI = 15 (each virus). To activate TRE and drive mCheery-T2A-$Na_v$1.1-expression, Doxycycline was added to the neuronal culture at 0.1 µg/ml, seven days after viral transduction. mCherry expression was detected after another seven days (*Figure 6C*).

A second group of lentiviruses encoding Dlxi1/2b-GFP or CMV-tRFP-$Na_v$1.1-shRNAs (scrambled, shRNA1, or shRNA2) were prepared by transfecting HEK293T cells and harvesting viral supernatants 36 hr after transfection. The helper plasmids used for Dlxi1/2b-GFP were pVSV-g, pRSVr and

pMDLg-pRRE (*Chen et al., 2013*). The helper plasmids for CMV-Na$_v$1.1-shRNAs were pCMV-dR8.2 dvpr and pCMV-VSV-g. The Dlxi1/2b-GFP virus was further concentrated by ultracentrifugation, while the CMV-tRFP-Na$_v$1.1-shRNAs viruses were used in the form of fresh supernatants. Tests of transduction efficiency showed that the amount of viruses produced from 1 cm$^2$ of 80% confluent HEK293T cells can be used to transduce 400,000 differentiating human neurons.

A third group of lentiviruses encoding TRE-mCherry-T2A-Na$_v$1.1-F383S, TRE-mCherry-T2A-Na$_v$1.1-F383S-S1328P, or EF1α-rtTA were prepared by transfecting HEK293N cells and harvesting viral supernatants 36 hr after transfection. The helper plasmids used were psPAX2 and pMD2.G. Viral supernatants were concentrated using Lenti-X Concentrator (Clontech, Mountain View, California) and viral titers were determined with a Lenti-X qRT-PCR Titration Kit (Clontech).

## NgN2-mediated neural differentiation of hESC line H9

H9 hESCs with stably integrated doxycycline-inducible Ngn2 and the tet-transactivator were maintained in mTESR medium on matrigel. One day prior to differentiation, cells were plated at 80,000 cells per well of a 6-well dish on hESC-qualified matrigel (Corning). The next day (day 0), medium was changed to differentiation medium (DMEM/F12, glutamax, N2 supplement, non-essential amino acids, penicillin/streptomycin, 10ng/mL BDNF, 10ng/mL NT3, and 2 µg/mL doxycycline [Dox]). On day two, the medium was changed; this time and thereafter the medium included B-27 supplement. On day three, cells were dissociated with accutase for 5 min, centrifuged for 5 min at 300 g, resuspended, and either 1 × 10$^6$ cells were seeded per well of a 6-well plate coated with matrigel or 1 × 10$^5$ cells were seeded per well of a 24-well plate containing matrigel-coated coverslips pre-seeded with rat astrocytes (Lonza). On day four, the media was changed and the dox concentration was reduced to 1 µg/mL. Partial media changes were performed on days six and eight to reduce the dox concentration to 0.5 µg/mL and 0.1 µg/mL, respectively. On day nine, cells were co-infected with equal titers of lentiviruses encoding EF1α-rtTA (3.0× 10$^8$ copies/cm$^2$ of culture area) and equal titers of lentiviruses (1.5 × 10$^8$ copies/cm$^2$ of culture area) encoding mCherry-T2A-Na$_v$1.1-F383S or mCherry-T2A-Na$_v$1.1-F383S-S1328P, and the dox concentration was raised to 2 µg /mL in half the wells. 50% media changes were done every other day with 2 µg /mL or 0.1 µg /mL dox until day 17, when the cells were lysed for western blotting or subjected to electrophysiology analysis.

## Immunofluorescence

All specimens were fixed at room temperature in 4% paraformaldehyde in PBS. The blocking buffer contained 10% normal goat serum, 3% bovine serum albumin, and 0.25% Triton-X 100 in PBS. The primary antibodies are listed in *Supplementary file 4*. The secondary antibodies were goat anti-rabbit/rat/mouse IgG conjugated with Alexa-488, −568 or −647 (Invitrogen). Images of cultured neurons were captured using a Zeiss M1 Axioscope; images of rat hippocampal slices were captured using a Zeiss spinning disc confocal microscope, equipped with 488/594/640 lasers (Perkin Elmer).

## Quantitative RT-PCR

Total RNA from samples of interest was purified using the RNeasy kit (Qiagen) and the RNase-free DNase Set (Qiagen). After normalization for concentration, RNA was reverse-transcribed into cDNA using the SuperScriptIII First-strand Synthesis SuperMix and random hexamer primers (Invitrogen). Real-time PCR for the majority of the cDNA samples was performed using the FastStart Universal SYBR Green Master (Rox) mix (Roche) on a Mastercycler realplex machine (Eppendorf) or a ViiA 7 machine (Applied Biosystems). A subset of the RNA samples were obtained at a later stage of the study, and the resulting cDNAs were analyzed by a 96.96 Dynamic array (Fluidigm, South San Francisco, California) and the EvaGreen detection method as described previously (*Paşca et al., 2011*). This subset includes cDNAs for the neuronal cultures from the control iPSC line 8858–3 and Dravet iPSC line 6358–2, as well as cDNAs for HEK293T cells used to assess Na$_v$1.1-shRNAs. All real-time PCR data were quantified relative to a reference gene by the delta-delta Ct method. All PCR primers are listed in *Supplementary file 3*.

## Single-cell RT-PCR to profile inhibitory neurons

At the end of a patch clamp recording, the cell soma was collected by a strong suction into the glass pipette, which contained 5 µL of internal solution. The pipette content was expelled into a 0.2 ml

PCR tube, which has been prefilled with 9 µl of a pre-amplification mix, and immediately frozen on dry ice. The pre-amplification mix contains multiplexed primers for genes of interest (55 nM, each oligo) and components of the CellsDirect One-Step qRT-PCR kit (Invitrogen). Cell samples were then processed on a PCR thermocycler, during which cellular RNAs were converted into cDNAs, and the cDNAs of genes of interest were pre-amplified selectively by gene-specific primers. Pre-amplified samples were treated with ExoSAP-IT (Affymetrix) to remove residual primers and were diluted four fold with water. Realtime PCR reactions were performed on a 96.96 Dynamic array (Fluidigm) using the EvaGreen detection method, as previously described (*Paşca et al., 2011*). An amplicon dissociation step (melting curve analysis) was included to determine PCR specificity. Data were collected and analyzed using Fluidigm Real-Time PCR Analysis software. A reaction with its threshold cycle ($C_t$) below 30 and a verified melting curve indicates the presence of expression for the gene of interest. All PCR primers are listed in *Supplementary file 3*.

## Assessing gene knockdown efficiency for $Na_v1.1$-shRNA lentiviruses

HEK293T cells were transduced with freshly produced lentiviral supernatants expressing CMV-tRFP-$Na_v1.1$-shRNAs (scrambled, shRNA1, and shRNA2). The viral transduction was carried out twice on two sequential days, with 6 hr of incubation in viral supernatants each day. Four days after the second transduction, the tRFP reporter was visible in the majority of cells for all three lentiviruses (*Figure 6—figure supplement 2A*), indicating sufficient and similar levels of expression among the three shRNA constructs. At this point the cells were transfected with a human $Na_v1.1$ expression plasmid pCAG-$hNa_v1.1$-IRES-EGFP. Three days later, total RNA was isolated and quantitative RT-PCR was performed on a 96.96 Dynamic array (Fluidigm) by the EvaGreen detection method, as previously described (*Paşca et al., 2011*). The pCAG-$hNa_v1.1$-IRES-EGFP plasmid was generated by inserting human $Na_v1.1$ cDNA (provided by GlaxoSmithKline and under MTA) into the NotI site of the pCA-GIG vector (Addgene Cat. 11159).

## Patch clamp electrophysiology and pharmacology

Patch clamp recordings were performed at room temperature using an EPC10 amplifier (HEKA, Germany), and signals were sampled at 10 kHz with a 3 kHz Bessel filter. The voltage-dependent sodium currents ($I_{Na}$) were recorded in the voltage clamp mode with P/8 leak subtraction. Traces acquired under a series resistance greater than 25 MΩ were discarded from analysis. Voltage protocols were applied to evaluate voltage-dependent activation and steady-state inactivation. Protocols details are described in *Figure 4—figure supplement 1*. The external solution for $I_{Na}$ contained: 50 mM NaCl, 94 mM TEA-Cl, 10 mM HEPES, 1 mM $CaCl_2$, 2 mM $MgCl_2$, 0.3 mM $CdCl_2$, 1 mM $BaCl_2$, 20 mM Glucose, and pH was titrated to 7.4. The internal solution contained 24 mM CsCl, 110 mM Cs-Methanesulfonate, 4 mM Mg-ATP, 0.3 mM $Na_2$-GTP, 10 mM $Na_2$-Phospocreatine, 10 mM EGTA, 10 mM HEPES, and pH was titrated to 7.3. The predicted liquid junction potential is -12 mV. Data reflecting voltage-dependent activation (i.e. the I-V relationship) were fitted with an equation in the form of $I = g*(V-V_R)/(1+exp(-0.03937*z*(V-V_{1/2})))$, where I is the amplitude of the current peak, g is a factor reflecting the number of active channels in the cell recorded, V is the membrane potential under which the current develops, z is the apparent gating charge, and $V_{1/2}$ is the half-maximal activation potential. Data reflecting steady-state inactivation were first normalized to the peak current under minimal inactivation (the first sweep) and then fitted with an equation in the form of $I = 1/(1+exp((V- V_{1/2}')/a'))$, where I is the normalized amplitude of the current peak, V is the voltage value of the inactivating pre-pulse, $V_{1/2}'$ is the half-maximal inactivating voltage and a' is a slope factor. Data describing the normalized channel conductance ($g/g_{max}$) of $Na_v1.1$-F383S and $Na_v1.1$-F383S-S1328P were fitted with $g/g_{max} = 1/(1+exp((V-V_{1/2})/a))$ (for voltage-dependent activation) and $g/g_{max} = 1/(1+exp((V- V_{1/2}')/a'))$ (for steady-state inactivation), where $V_{1/2}$ and $V_{1/2}'$ are half-effective membrane potentials, and a and a' are slope factors. Neuronal action potentials were recorded in the current clamp mode. Resting membrane potential was recorded immediately after the pipette breaks in. Then neurons were held at –65 mV, and sustained current injections (duration, 1 s; increment, 10 pA) were applied to evoke action potentials. The external solution contained: 140 mM NaCl, 2.5 mM KCl, 2.5 mM $CaCl_2$, 2 mM $MgCl_2$, 1 mM $NaH_2PO4$, 10 mM HEPES, 20 mM Glucose, and pH was titrated to 7.4. The internal solution contained 120 mM K-Gluconate, 20 mM KCl, 4 mM NaCl, 4 mM Mg-ATP, 0.3 mM $Na_2$-GTP, 10 mM $Na_2$-Phospocreatine, 0.5 mM

EGTA, 10 mM HEPES, and pH was titrated to 7.25. The predicted liquid junction potential is -14.4 mV. Tetrodotoxin citrate (TTX) was solubilized in water as 10 mM stocks and used at 0.8 µM or 1 µM in external solution. Gabazine (SR 95,531 hydrobromide, Tocris) was solubilized in DMSO as 10 mM stocks and used at 10 µM in external solution. Data were initially analyzed with the Patchmaster software (HEKA). Further analysis was performed using IgorPRO, Microsoft Excel, and GraphPad Prism. The iPSC lines involved in each electrophysiology experiment are described in *Supplementary file 2*.

## Western blotting

Human NgN2-induced neurons were maintained in 6-well plates until day 17. Membrane proteins were extracted using the Mem-PER Plus kit (ThermoFisher Scientific), with all processing steps carried out on ice or at 4°C and the permeabilization and solubilization buffers supplemented with Halt protease and phosphatase inhibitor cocktail (ThermoFisher Scientific). Solubilized membrane protein samples were quantified with Pierce BCA Protein Assay kit (Thermofisher Scientific). Quantity-normalized samples were supplemented with the 4x NuPAGE LDS sample buffer (ThermoFisher Scientific) and 2% (v/v, final concentration) β-mercaptoethanol and incubated at 70°C for 5 min. The samples (12.5 µg of total membrane protein) were then loaded to a Novex 3–8% Tris-Acetate gel (ThermoFisher Scientific), together with Chameleon Duo pre-stained protein ladder (LI-COR, Lincoln, Nebraska). After gel electrophoresis, proteins were transferred to a nitrocellulose membrane using the iBlot2 system (ThermoFisher Scientific). The high-molecular-weight (above 160 kDa) half of the membrane was blotted with a $Na_v1.1$ antibody (AB5204/RRID AB_91751, EMD Millipore; used at 1:200), and the low-molecular-weight (below 125 kDa) half was blotted with an antibody for Na/K-ATPase (ab 76020/RRID AB_1310695, Abcam, Cambridge, Massachusetts), a ubiquitous membrane protein serving as the sample loading control. Detection of both proteins was carried out with a HRP conjugated secondary antibody (W401B, used at 1:5000, Promega, Madison, Wisconsin) and the SuperSignal West Femto Chemiluminescent Substrate (ThermoFisher Scientific) on a ChemiDoc MP imaging system (BioRad).

## Acknowledgements

The authors are grateful to the family who participated in this study. We thank Jennifer M Perry (clinical research associate) for her assistance in enrolling the family. Philip Manos, Dr. Arnaud Lacoste, and Dr. YounKyoung Lee kindly shared control iPSC line WT-GM08402A-SeV-hiPSC 1, which we used to generate the subclone 8402-2 WT9. WT-GM08402A-SeV-hiPSC 1 was derived from the fibroblast cell line GM08402 from the NIGMS Human Genetic Cell Repository at the Coriell Institute for Medical Research. We thank Maria L Fabian and Dr. Ben A Barres for providing rat cortical astrocytes, Dr. Lief Fenno and Dr. Karl Deisseroth for the pLenti-Syn1-ChR2(H134R)-YFP virus, and Dr. Michael Lochrie and the Stanford University Neuroscience Gene Vector core for generating and titering other lentiviral vectors. The wild-type human $Na_v1.1$ cDNA described in *Figure 6* was acquired from GlaxoSmithKline (GSK) and used in the Dolmetsch lab at Stanford University under an MTA between GSK and Stanford. The authors acknowledge Dr. Massimo Mantegazza, Dr. Andrew Powell, Dr. Mike Bird, Dr. Jane Lewis, and Dr. Jaysen Rajkomar for their help with material transfer. The $Na_v1.1$ cDNAs encoding p.F383S and p.F383S-p.S1328P mutations was synthesized de novo at Genewiz and Novartis with help from Dr. Paul Groot-Kormelink. We also thank Dr. Alex Shcheglovitov and Dr. Jason Bant for their advice on electrophysiology techniques, as well as Dr. Masayuki Yazawa, Dr. Ajamete Kaykas, Dr. Shengda Lin, Jesse Mull, Marie Sondey, Erica Seigneur, David M Lipton, Le An, Alex Sosa, Samantha Madala, Kimberly Hsu, and members of the Dr. David Prince's lab and Dr. John Huguenard's lab for scientific discussions and assistance with various lab techniques. This work was supported by private funds from the Hsiao family (to YS), Stanford University School of Medicine Dean's postdoctoral fellowship (to YS), the Epilepsy Foundation postdoctoral fellowship (to YS), the Swiss National Science Foundation postdoctoral fellowship (PBSKP3-123434, to TP), Grants MH049428, RB2-1602 from California Institute for regenerative Medicine (CIRM) to JLRR, Grant RT2-01906 from CIRM to RED.

# Additional information

## Competing interests

YS, CG, KAW, WG, KC, RED: Currently employed by the Novartis Institutes for BioMedical Research Inc. TP: Currently employed by Circuit Therapeutics Inc. Y-JJC: Currently employed by Genentech, a member of the Roche Group. The other authors declare that no competing interests exist.

## Funding

| Funder | Grant reference number | Author |
|---|---|---|
| Epilepsy Foundation | Postdoctoral fellowship | Yishan Sun |
| California Institute for Regenerative Medicine | Grant RT2-01906 | Ricardo E Dolmetsch |
| Schweizerischer Nationalfonds zur Förderung der Wissenschaftlichen Forschung | Postdoctoral fellowship PBSKP3-123434 | Thomas Portmann |
| Stanford University School of Medicine | Dean's postdoctoral fellowship | Yishan Sun |
| California Institute for Regenerative Medicine | MH049428 | John LR Rubenstein |
| Private funds from the Hsiao Family | Postdoctoral research fund | Yishan Sun |
| California Institute for Regenerative Medicine | RB2-1602 | John LR Rubenstein |

The funders had no role in study design, data collection and interpretation, or the decision to submit the work for publication.

## Author contributions

YS, TP, KAW, Conception and design, Acquisition of data, Analysis and interpretation of data, Drafting or revising the article; SPP, Conception and design, Acquisition of data, Analysis and interpretation of data, Drafting or revising the article, Contributed unpublished essential data or reagents; CG, WG, KCC, HG, Acquisition of data, Analysis and interpretation of data, Drafting or revising the article; DV, JLRR, Analysis and interpretation of data, Drafting or revising the article, Contributed unpublished essential data or reagents; Y-JJC, Analysis and interpretation of data, Contributed unpublished essential data or reagents; RM, Conception and design, Acquisition of data; KC, Acquisition of data, Analysis and interpretation of data; DVM, JH, Analysis and interpretation of data, Drafting or revising the article; WMF-S, JAB, RED, Conception and design, Analysis and interpretation of data, Drafting or revising the article

## Author ORCIDs

Ricardo E Dolmetsch, http://orcid.org/0000-0002-2738-8338

## Ethics

Human subjects: All experiments involving primary fibroblasts and iPSCs from the Dravet Syndrome patients and healthy subjects were carried out at Stanford University after informed consent and under study protocols (No. 12481, No. 232, and No. 327 ) approved by the institutional review board (IRB) and Stem Cell Research Oversight (SCRO) of Stanford University. For drafting and revising the manuscript, authors currently affiliated with Novartis (Y.S., C.G., K.A.W., W.G.,K.C.,and R.E. D.) analyzed iPSC-related data that had been previously generated at Stanford University. They also incorporated new data from a control iPSC line (8402-2 WT9) that was generated at Novartis from fibroblast cell line GW08402, in accordance with the conditions of the NIGMS Repository Samples, governed by the Coriell IRB in accordance with DHHS regulations, as outlined in the MTA between Coriell and Novartis.

Animal experimentation: In this study primary cultures of rat cortical neurons, rat cortical astrocytes, and rat hippocampal organotypic slices were prepared in strict accordance with lab protocols granted to the Novartis neuroscience department (No. 14 NSC 024), the lab of Dr. Ben A. Barres

(No. 10726), and the lab of Dr. Daniel V. Madison (No. 10565), respectively, under the governance of the institutional animal care and use committee (IACUC) of Novartis and the Administrative Panel on Laboratory Animal Care (APLAC) of Stanford University.

## Additional files

### Supplementary files

• Supplementary file 1. SNP array assessment of copy number variants (CNVs) in iPSCs.

• Supplementary file 2. Pluripotent stem cell lines characterized by electrophysiology.

• Supplementary file 3. qRT-PCR and single-cell RT-PCR primers.

• Supplementary file 4. Primary antibodies used for immunofluorescence.

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
