## [Decision Letter]

Thank you for submitting your work entitled "Deficit in inhibitory neuron activity and amelioration by Cannabidiol in a human cellular model of Dravet Syndrome" for consideration by *eLife*. Your article has been reviewed by two peer reviewers, and the evaluation has been overseen by William A. Catterall as the Reviewing Editor and Gary Westbrook as the Senior Editor. One of the two reviewers involved in the review of your submission has agreed to reveal their identity: Reviewer, Scott Baraban.

The reviewers have discussed the reviews with one another and the Reviewing Editor has drafted this decision to help you prepare a revised submission.

Dravet Syndrome is an intractable genetic epilepsy syndrome with substantial premature death and severe co-morbidities, which are caused by mutations in the *SCN1A* gene encoding Na_v_1.1 voltage-gated brain sodium channel. Induced pluripotent stem cell methods are used here to determine the functional impact of a Dravet Syndrome mutation. Previous published work on iPSC-derived neurons has given inconsistent results. The highlights of this work are: 1. well-developed and well-defined iPSC-derived neurons; 2. evidence that a Dravet Syndrome mutation impairs the sodium currents and excitability of inhibitory neurons; 3. evidence that excitatory pyramidal cells are unaffected; and 4. evidence that the nonpsychotropic marijuana component cannabidiol inhibits excitability of excitatory neurons and increases excitability of inhibitory neurons.

Summary and essential revisions:

The recommendation of the reviewers and guest editor is for major revisions and resubmission.

1) The work on cannabidiol is considered to be premature and tangential to the main thrust of this manuscript, and therefore we recommend that it be developed as a separate manuscript for submission elsewhere.

2) The reviewers and guest editor are also concerned that, as now written, the manuscript seems relevant to only a single Dravet Syndrome mutation out of nearly one thousand. However, the significance of the work could be substantially enhanced by demonstrating that this mutation is a complete, or nearly complete, loss-of-function mutation, because the majority of Dravet Syndrome mutations are thought to be complete loss-of-function. This study could therefore serve as a model for this large set of disease mutations. This question could be addressed by introducing a single-residue mutation to block tetrodotoxin inhibition, expressing the mutant and an appropriate control in your iPSC-derived neurons, blocking endogenous sodium channels with tetrodotoxin, and determining the functionality of the toxin-resistant Dravet Syndrome mutant. Several groups have used this approach successfully in other contexts.

3) An additional point of interest is to determine whether this mutant is expressed as a protein at all, in order to assess whether dominant negative effects of the mutant on wild-type sodium channels are possible through association with a limiting supply of auxiliary subunits or through other means.

4) Interpretation of these interesting findings, in the context of an intact network, requires a more complete understanding of the precise interneuron identity. Although this may seem a bit trivial at first glance, it is critical to how the two major findings presented here are interpreted. We have learned much about interneuron diversity and interneuron origins since the initial 2006 Catterall et al. identification of a defect in firing associated with a GABA-expressing bipolar neuron from Na_v_1.1 mutant mice. Rubenstein, Fishell, Marin and many others have carefully classified GABA-expressing neurons into a wide variety of sub-populations with distinct embryonic origins. From this literature, and looking at the data presented here – putative interneurons with a telencephalic identity express GABA or GAD (30-50%), and of these 50% express calretinin with a very small percentage expressing SST and none expressing PV. This would suggest that the iPS protocol used enriches for GABA-expressing CR+ interneurons and would be more in line with a Caudal (not Medial) Ganglionic Eminence origin designation; current-clamp firing properties shown in Figure 3 are also consistent with a CR+ Regular-Spiking Non-Pyramidal cell designation. Interneurons derived from the MGE are primarily parvalbumin+ (~65%) and somatostatin+ (~35%) sub-types and it is rare to have CR+ interneurons derived from an MGE lineage (Anderson et al. Cereb. Cortex 2002; Xu et al. J. Neurosci. 2004; Butt et al. Neuron 2005; Fishell et al. Novartis Found. Symp. 2007; Gelman et al. Jaspers 2012). It would be helpful to know if the interneurons shown here also express 5HT3a receptors (see below), VIP or CoupTFII, as this would further help in distinguishing interneuron sub-populations.

5) If the telencephalic interneurons shown here are primarily CGE-derived CR+ interneurons, an interneuron cell population that can target other interneurons (Freund and Buzsaki, Hippocampus 1996; Gonchar and Burkhalter, Cereb. Cortex 1999; Chamberland et al. Front. Cell Neurosci. 2010; Urban et al. Acta Biol. Hung. 2002), then the interpretation that DS-derived interneurons have reduced sodium current and reduced firing would functionally result in a loss of inhibitory tone onto inhibitory neurons or an overall disinhibition placed in the context of a compete network environment. This would not be a reasonable explanation for epilepsy in DS, as it would be associated with increased network excitability and likely an increased propensity toward seizures. Nor would this be entirely consistent with mouse data from Catterall and other groups who have demonstrated that Cre-mediated inactivation of Na_v_1.1 in SST+ or PV+ interneuron sub-populations (which primarily target excitatory neurons) can result in epileptic phenotypes.

6) Firing properties depicted are not entirely consistent with those expected for mature interneurons or principal cells. First, the representative traces in Figure 5 do not match very well with the quantitative input-output plots. The excitatory neuron appears to exhibit spike frequency accommodation, as expected in the plot but not the sample traces. Whereas the control interneuron, which would not be expected to show SFA does appear to accommodate. Second, the spike firing rates shown for interneurons and excitatory neurons appear equivalent. But mature interneurons should have much higher firing frequencies.

---

## [Author Response]

Summary and essential revisions:

*The recommendation of the reviewers and guest editor is for major revisions and resubmission.*

1) The work on cannabidiol is considered to be premature and tangential to the main thrust of this manuscript, and therefore we recommend that it be developed as a separate manuscript for submission elsewhere.

We thank our reviewers and guest editor for this recommendation to refine the focus of the paper. In the revised manuscript we have removed the text and figures describing cannabidiol.

2) The reviewers and guest editor are also concerned that, as now written, the manuscript seems relevant to only a single Dravet Syndrome mutation out of nearly one thousand. However, the significance of the work could be substantially enhanced by demonstrating that this mutation is a complete, or nearly complete, loss-of-function mutation, because the majority of Dravet Syndrome mutations are thought to be complete loss-of-function. This study could therefore serve as a model for this large set of disease mutations. This question could be addressed by introducing a single-residue mutation to block tetrodotoxin inhibition, expressing the mutant and an appropriate control in your iPSC-derived neurons, blocking endogenous sodium channels with tetrodotoxin, and determining the functionality of the toxin-resistant Dravet Syndrome mutant. Several groups have used this approach successfully in other contexts.

We agree that characterizing the p.S1328P mutation is important for this paper so we have done these experiments and have added them as a new Figure 1 in the paper. In the revised manuscript we expressed a TTX-resistant version (Na_v_1.1-F383S-S1328P) of Na_v_1.1 in human neurons and compared it to a control lacking the S1328P mutation. We expressed the channel in NgN2-induced, hESC-H9-derived human telencephalic neurons and found that while the mutant channel is expressed at similar levels to the control channel it has defects in trafficking that result in reduced expression of functional channels. In addition, has altered gating properties including a hyperpolarizing shift of the steady-state inactivation curve of 15 mV and a minor hyperpolarizing shift of the activation voltage of the channel of 5 mV. On balance, the impairment in trafficking and the increased inactivation results in a channel that is less active particularly during trains of action potentials when the fraction of inactivated channels is likely to accumulate. This in fact is what we observed in the patient cells. These new findings are presented in text (subsection “The Na_v_1.1-p.S1328P mutation impairs functional expression and alters voltage-dependent gating of the channel”), Figure 1, Figure 1—figure supplement 1, Table 3, and Table 4 in the new manuscript.

We were also able to perform a study using rat cortical neurons and the same expression constructs. The findings are very similar to those in human NgN2 neurons. This data is somewhat redundant with what is shown in in the manuscript using NGN2 human cortical neurons so we are not including it in the manuscript but are including it as Figure 7 for the reviewers.

Author response image 1.**DOI:**
http://dx.doi.org/10.7554/eLife.13073.039

3) An additional point of interest is to determine whether this mutant is expressed as a protein at all, in order to assess whether dominant negative effects of the mutant on wild-type sodium channels are possible through association with a limiting supply of auxiliary subunits or through other means.

Thank you for raising this point. As described in our response to comment #2, Na_v_1.1-F383S and Na_v_1.1-F383S-S1328P are expressed at similar levels in the NgN2 neurons by western blot, indicating normal biogenesis (protein synthesis). Therefore it is possible that the p.S1328P mutant may have a dominant-negative effect on wild-type Na_v_1.1. At first glance it doesn’t appear that the p.S1328P channel is dominant negative because we can measure significant sodium currents in the patient neurons but we can’t discount a subtle dominant negative effect of the mutation.

4) Interpretation of these interesting findings, in the context of an intact network, requires a more complete understanding of the precise interneuron identity. Although this may seem a bit trivial at first glance, it is critical to how the two major findings presented here are interpreted. We have learned much about interneuron diversity and interneuron origins since the initial 2006 Catterall et al. identification of a defect in firing associated with a GABA-expressing bipolar neuron from Nav1.1 mutant mice. Rubenstein, Fishell, Marin and many others have carefully classified GABA-expressing neurons into a wide variety of sub-populations with distinct embryonic origins. From this literature, and looking at the data presented here – putative interneurons with a telencephalic identity express GABA or GAD (30-50%), and of these 50% express calretinin with a very small percentage expressing SST and none expressing PV. This would suggest that the iPS protocol used enriches for GABA-expressing CR+ interneurons and would be more in line with a Caudal (not Medial) Ganglionic Eminence origin designation; current-clamp firing properties shown in Figure 3 are also consistent with a CR+ Regular-Spiking Non-Pyramidal cell designation. Interneurons derived from the MGE are primarily parvalbumin+ (~65%) and somatostatin+ (~35%) sub-types and it is rare to have CR+ interneurons derived from an MGE lineage (Anderson et al. Cereb. Cortex 2002; Xu et al. J. Neurosci. 2004; Butt et al. Neuron 2005; Fishell et al. Novartis Found. Symp. 2007; Gelman et al. Jaspers 2012). It would be helpful to know if the interneurons shown here also express 5HT3a receptors (see below), VIP or CoupTFII, as this would further help in distinguishing interneuron sub-populations.

Thank you for this suggestion. We performed additional studies to characterize the iPSC-derived inhibitory inter-neurons generated using our protocol. We would like to summarize our interpretation of the literature as well as what we found in the additional studies:

A) Nicholas et al. (Cell Stem Cell, 2013, PMID 23642366) demonstrated that human iPSC/ESC-derived MGE progenitors produce up to ~80% Calretinin positive GABAergic neurons in the time frame of their neural differentiation (up to 30 weeks). The authors also showed that human fetal MGE explants produced prevalent Calretinin-positive GABAergic neurons. Importantly, the authors demonstrated co-expression of Calretinin and the MGE marker NKX2.1. Specifically, Calretinin expression and the NKX2.1-GFP reporter overlapped in hiPSC/ESC derived GABAergic neurons and Calretinin staining was also found in the 15-gestational-week human fetal MGE. Although it is believed that in the adult mammalian brain Calretinin expression is largely restricted to CGE-derived interneurons, it appears that human interneurons go through a stage in which they express calretinin. To examine this possibility we performed co-immunostaining for Calretinin and NKX2.1 in our inhibitory neuron cultures at roughly half way of the differentiation protocol, we found many NKX2.1-positive progenitor-like cells co-express Calretinin, and we also identified cells with neuronal morphology that co-express both. These results are described in Figure 3—figure supplement 1 and in text (subsection “Verifying identity of iPSC-derived inhibitory neurons”, second paragraph).

B) Coup-TFII and Sp8 are transcription factors that are strongly expressed in CGE derived neuronal progenitors. We found that Coup-TFII and Sp8 were only weakly expressed in the neural stem cells/progenitors that form the inhibitory neurons in our study. By quantitative RT-PCR, the neural rosettes (containing neural stem cells/progenitors) generated from the ventral telencephalon patterning protocol showed a strong expression of the MGE-marker NKX2.1 but only mild induction of CoupTF-II and Sp8. These results are described in Figure 2 and Figure 2—figure supplement 2 panel C and in text (subsection “Verifying identity of iPSC-derived inhibitory neurons”, first paragraph).

C) Only less than 6% of the inhibitory neurons in our study express the CGE-interneuron-specific marker 5HT3aR. We performed the analysis by single-cell RT-PCR using cytoplasmic RNA harvested with the patch-clamp pipette. 5HT3aR was only found in 0.9% of the inhibitory neurons derived from control ESCs and iPSCs and 5.4% of the inhibitory neurons derived from Dravet iPSCs. In this analysis we also saw low-percentage expression of VIP (<10%), CCK(<20%) and NPY (<20%), and ~30% of the inhibitory neurons express Reelin. Although VIP, CCK,NPY and Reelin are commonly found in CGE derived interneurons, they are also expressed in sub-populations of MGE-derived interneurons. Overall, our data suggest only a small fraction of inhibitory neurons are of CGE origin. Please see Figure 3—figure supplement 2 and text (subsection “Verifying identity of iPSC-derived inhibitory neurons”, third paragraph) in the revised manuscript.

In summary, our data suggest that the majority of the inhibitory neurons in our study represent an MGE identity.

5) If the telencephalic interneurons shown here are primarily CGE-derived CR+ interneurons, an interneuron cell population that can target other interneurons (Freund and Buzsaki, Hippocampus 1996; Gonchar and Burkhalter, Cereb. Cortex 1999; Chamberland et al. Front. Cell Neurosci. 2010; Urban et al. Acta Biol. Hung. 2002), then the interpretation that DS-derived interneurons have reduced sodium current and reduced firing would functionally result in a loss of inhibitory tone onto inhibitory neurons or an overall disinhibition placed in the context of a compete network environment. This would not be a reasonable explanation for epilepsy in DS, as it would be associated with increased network excitability and likely an increased propensity toward seizures. Nor would this be entirely consistent with mouse data from Catterall and other groups who have demonstrated that Cre-mediated inactivation of Na_v_1.1 in SST+ or PV+ interneuron sub-populations (which primarily target excitatory neurons) can result in epileptic phenotypes.

We thank the reviewers and guest editor pointing this out as a caveat. As pointed out above, we believe that the neurons we are generating are largely MGE derived and not CGE derived. In addition it appears that not all CGE derived CR+ interneurons target interneurons. For example, Gonchar and Burkhalter (Cereb. Cortex, 1999) found that “in (visual cortex) layers 5 and 6, 60% of CR+ terminals form synapses with GABA negative somatic profiles”. Urban et al. (Acta Biol. Hung. 2002) found that “(in the hippocampal CA1 subfield, one type of CR positive terminals) formed symmetric synapses on both pyramidal and interneuron dendrites. Distribution of postsynaptic targets showed that 26.8% of the targets were CR positive interneuron dendrites, and 25.2% proved to be proximal pyramidal dendrites. CR negative interneuron dendrites were also contacted (12.4%).” Having said this, the only convincing way to answer whether the human neurons that we are generating in our system target excitatory or inhibitory neurons and affect network excitability is to study them in a network that reflects the relative numbers of excitatory and inhibitory neurons in the human cortex. This is still a difficult experiment to do with the current state of the technology.

6) Firing properties depicted are not entirely consistent with those expected for mature interneurons or principal cells. First, the representative traces in Figure 5 do not match very well with the quantitative input-output plots. The excitatory neuron appears to exhibit spike frequency accommodation, as expected in the plot but not the sample traces. Whereas the control interneuron, which would not be expected to show SFA does appear to accommodate. Second, the spike firing rates shown for interneurons and excitatory neurons appear equivalent. But mature interneurons should have much higher firing frequencies.

Because human neurons in culture have a prolonged maturation time, we don’t think that the neurons that we have generated are completely mature. We have found that at this stage in development, inhibitory interneurons in culture do not have the fast firing rate of mature inhibitory neurons. Others in the field have demonstrated their prolonged maturation time course that roughly matches the timing of human embryonic brain development (Nicholas et al. Cell Stem Cell, 2013; Maroof et al. Cell Stem Cell, 2013) and our observations agree with these findings. In the revised manuscript, we discuss the limited time frame of in vitro differentiation (subsection “Enrichment of Na_v_1.1 in inhibitory neurons correlates with its contribution to neuronal excitability” and Discussion, last paragraph) but do not describe the neurons as mature.

Spike frequency accommodation (SFA) is indeed a common feature of the developing inhibitory interneurons in our study. We examined current clamp traces for all inhibitory interneurons derived from control ESC and iPSC lines, and we found that 73.3% have variable degrees of SFA, while small fractions of the neurons show non-accommodating (3.3%) or stuttering (5%) patterns. The remaining fraction (18.4%) fired immature or single action potentials, consistent with the nature of the neuronal culture being “developing in a culture dish”. In our opinion, the prevalence of the accommodating pattern is mostly reflecting the developmental timing rather than cell type or lineage, although it is worth noting that in the adult mouse brain, accommodating spike patterns are actually found to various degrees in subpopulations of Martinotti cells, bipolar cells, bitufted cells, double bouquet cells, and even basket cells and Chandelier cells (Markram et al. Nature Reviews Neuroscience, 2004; Figure 6).

Regarding the maximal spike firing rates, we did not see a statistical difference between the inhibitory neurons and excitatory neurons derived from control iPSC/ESC lines, although we are seeing a few inhibitory neurons firing at near or above 30 Hz (See the graph in Figure 8 on the left). Again, such a spike frequency profile could reflect the developmental timing of the neurons. Interestingly, we could observe a difference in the half-width of excitatory neurons and inhibitory interneurons derived from the same control iPSC line (e.g. line 6593-8, see the graph in Figure 8 on the right).

Author response image 2.**DOI:**
http://dx.doi.org/10.7554/eLife.13073.040